# Drivers and impact of the early silent invasion of SARS-CoV-2 Alpha

Benjamin Faucher[1], Chiara E. Sabbatini[1], Peter Czuppon [2],
Moritz U. G. Kraemer[3,4], Philippe Lemey [5], Vittoria Colizza [1,6,9],
François Blanquart[7,9], Pierre-Yves Boëlle[1,9] & Chiara Poletto [8,9] ✉

SARS-CoV-2 variants of concern (VOCs) circulated cryptically before being identified as a threat, delaying interventions. Here we studied the drivers of such silent spread and its epidemic impact to inform future response planning. We focused on Alpha spread out of the UK. We integrated spatio-temporal records of international mobility, local epidemic growth and genomic surveillance into a Bayesian framework to reconstruct the first three months after Alpha emergence. We found that silent circulation lasted from days to months and decreased with the logarithm of sequencing coverage. Social restrictions in some countries likely delayed the establishment of local transmission, mitigating the negative consequences of late detection. Revisiting the initial spread of Alpha supports local mitigation at the destination in case of emerging events.

In December 2020, one year after SARS-CoV-2 emergence, the increased transmissibility and severity of the Alpha variant (Pango lineage B.1.1.7) prompted an international alert[1,2]. Attempts to contain the variant in the UK, where it was first identified, were too late and its global dissemination led to a resurgence of cases and deaths in many countries. Sequences shared through GISAID[3] in real time provided records of the variant's international spread[4] and a number of studies predicted the first countries that would be invaded based on international travel from the UK[5–7]. Still, observations were not in agreement with the expectations, and it soon became clear that the first Alpha detection in countries outside the UK occurred when the variant had been circulating silently in these territories for some time. For instance, the first case infected by the Alpha variant was identified on 25 Dec 2020 in France[3]; yet, three weeks later, already 3% of the ~100,000 weekly reported COVID-19 cases were caused by the Alpha lineage[8]. Late detection was also noted in Switzerland[9] and the USA[10,11].

Phylodynamics analysis and modeling studies revealed that silent spread occurred for early SARS-CoV-2 lineages and subsequent variants of concern (VOCs)[12–20]. This has sparked a public health debate. The efforts to contain a variant at the source are ineffective if they come too late, when the virus is already spreading cryptically out of the source. Interventions aiming at mitigation or delay may instead have an impact depending on the extent and duration of silent dissemination at the time they are implemented[21,22]. Recent works addressed the minimal sequencing coverage to detect a variant early enough for an effective response, and proposed modeling tools for risk assessment[23–27]. However, the complex interplay of factors determining the duration of silent propagation remains poorly understood. Indeed, SARS-CoV-2 VOCs emerged in a context of changing patterns of genomic surveillance, international travel, population immunity, and local interventions. When Alpha emerged in late 2020, sequencing coverage was highly variable and changed dramatically as countries increased genomic surveillance. It took months from the emergence to declaring Alpha a VOC[2]. During this period the epidemiological context across many regions changed substantially. The efforts in the UK and other countries to control a substantial autumn pandemic wave

[1]Sorbonne Université, INSERM, Institut Pierre Louis d'Epidémiologie et de Santé Publique (IPLESP), F75012 Paris, France. [2]Institute for Evolution and Biodiversity, University of Münster, Münster 48149, Germany. [3]Department of Biology, University of Oxford, Oxford, UK. [4]Pandemic Sciences Institute, University of Oxford, Oxford, UK. [5]Department of Microbiology, Immunology and Transplantation, Rega Institute, Laboratory for Clinical and Epidemiological Virology, KU Leuven, Leuven, Belgium. [6]Department of Biology, Georgetown University, Washington, DC, USA. [7]Center for Interdisciplinary Research in Biology, CNRS, Collège de France, PSL Research University, Paris 75005, France. [8]Department of Molecular Medicine, University of Padova, 35121 Padova, Italy. [9]These authors contributed equally: Vittoria Colizza, François Blanquart, Pierre-Yves Boëlle, Chiara Poletto. ✉e-mail: chiara.poletto@unipd.it

impacted the rate of exportations of Alpha out of the UK and the chance to seed local transmission. This makes the emergence of Alpha a paradigmatic example.

Here we used a Bayesian model to retrospectively reconstruct the initial international dissemination of Alpha from 20 Sep 2020 to 31 Dec 2020 out of the UK. By leveraging diverse sources of data for the temporal and geographical change in international travel, sequencing coverage and local epidemic growth, we show that these factors, together with the effect of the international VOC alert on surveillance, drove the duration of Alpha silent spread.

## Results

### Factors contributing to the spread of Alpha
The early spread of the Alpha variant in the UK occurred in the last quarter of 2020, in a context where a lockdown in the UK, from 5 Nov to 2 Dec 2020, reduced local transmission and the potential for international propagation[28–30]. Air, train, Channel Tunnel and ferry passengers traveling out of the UK in this month had fallen up to 20% of that in September (Fig. 1A).

Over the same period, more than 200,000 sequences were submitted to GISAID from 73 countries, which allowed monitoring the spread of Alpha. We defined the date of first Alpha detection in each country as the date of collection of the first Alpha sequence submitted to GISAID. We hypothesized that sequences collected earlier but submitted at a later date resulted from retrospective surveillance and would misrepresent the routine screening effort. Sequencing coverage ranged over four orders of magnitude over countries: 59% of the cases reported in New Zealand over Sep–Dec 2020 were sequenced, but the median for all countries was only at 0.3%. As might be expected, the date of first detection of Alpha was earlier with higher sequencing coverage and more travelers from the UK (Fig. 1B). The UK was the only country to report the Alpha strain before Dec 1, 2020, followed by Denmark (2 Dec 2020) and Australia (7 Dec 2020). The Alpha international alert on 18 Dec 2020, led to a rise in sequencing coverage (Fig. 1C), shorter collection-to-submission times for Alpha sequences than for others (27 days (CI [8,137]) vs. 52 days (CI [10,162]), Fig. 1D and Supplementary Fig. 1) and prioritization of sequencing of travelers from the UK[4,31]. Nineteen countries collected their first Alpha sequence the week following the alert and submitted it with a median delay of 9 days. In most of these countries, the first case detected was a case imported from the UK[32].

We developed the Alpha international dissemination model to fit the date of first detection and the corresponding date of submission between the beginning of September and end of December in the 73 countries contributing to GISAID during the period. We used dates for 24 countries where the Alpha was detected during the period (including the UK) and accounted for no detection in the other countries by statistical censoring. The key assumption of the model is that the hazard of submitting an Alpha sequence in a country outside the UK results from the dynamically changing incidence in the UK, outbound flows of travelers from the UK, sequencing coverage at arrival and the delay from collection to submission. Thus, we assumed that before the end of December, the first detected cases were traveling cases[4,32] and dissemination was at its early stage, i.e. traveling cases were traveling out of the UK. Although a simplification, this is in line with earlier work showing that the UK was the main source of Alpha dissemination during the first three months, while other countries became more important at a later stage[19]. Time-varying incoming travelers from the UK, sequencing coverage and collection-submission delays were derived from data for each country. Fitted parameters were the exponential growth rate in the UK before and after the beginning of the November lockdown and the increase in genomic surveillance among travelers compared to cases in the community in destination countries following the international alert. Details are given in the Methods section.

Observed dates of first detection and submission (Fig. 2A) and a cumulative number of countries submitting an Alpha sequence (Fig. 2B) matched the model predictions. Portugal and Germany detected Alpha earlier than predicted by our model; there the delays from collection to submission were the longest (48 days for Portugal and 23 days for Germany, versus a median of 9 days in the other countries submitting Alpha). For Portugal, the long gap between the collection dates of the first and the second submitted sequences suggests a retrospective investigation. The model predicted a median seeding date of the Alpha epidemic in the UK on 8 Sep 2020 (95% prediction interval [Aug 21, Sep 19])[33]. The estimated doubling time of incidence in the UK was 4.2 days (95% crI [3.6, 5.3]) before 5 Nov 2020 and 10.6 days (95% crI [6.5, 22]) afterwards. Assuming the reproductive ratio $R = 1 + rT$, with $T$ the generation time interval at 6.5 days[34] and $r$ the Alpha exponential growth in the UK, these estimates would be compatible with $R = 2.0$ [1.8, 2.3] and $R = 1.4$ [1.1, 1.65] before and after 5 Nov 2020. These values broadly agree with previous estimates, with a pattern of decreased transmission over time[28–30,33,35]. With these estimates, the predicted trend of Alpha infections in the UK was in agreement with the observations (Fig. 2C)[36]. The large number of countries reporting Alpha almost simultaneously in late December was explained by an estimated 50-fold (95% crI [12, 298]) increase of sequencing coverage among travelers compared to non-travel related cases following the alert, consistently with the active search of Alpha cases among travelers and their contacts. Further details on parameter estimates and fit convergence are reported in Supplementary Fig. 2 and Supplementary Table 5.

In a sensitivity analysis, results were found to be robust to a range of modeling assumptions—e.g. changepoints for the exponential growth of incidence in the UK, rate of detection of COVID-19 infections outside the UK, and incubation period among the others. Details are reported in Supplementary Table 5.

### Silent spread ranged from days to weeks
We next used the international dissemination model to predict the date of the first introduction of Alpha from the UK to each of the 73 countries under study and the duration of silent spread, i.e. the duration of the time from the first introduction to the first detection of Alpha. We found that up to ~65 countries could have experienced the introduction of Alpha by the end of December, compared with the 24 countries that reported it (Fig. 3A). Our model predicted that the first introduction of Alpha in a country occurred up to 70 days earlier than the date of first Alpha detection (Fig. 3B, C). For instance, our model predicted that Alpha arrived 60 days earlier in Italy with an average sequencing coverage of 0.3% during the period, while it was only 15 days in Hong Kong with a sequencing coverage of 50%. Overall, the duration of the silent spread showed a logarithmic association with the average sequencing coverage (Fig. 3D). The estimated dissemination pattern is consistent with real-time projections based on air-travel[5]. Early introductions in Denmark and the USA were also consistent with the result of phylodynamic analyses and retrospective surveillance[10,11,37–39]. We found that the collection date of the first Alpha sample ever collected in each country (earlier than the first detection in 34 countries because of retrospective surveillance) was within the range of first introduction predicted by the model but for Colombia.

### Local dynamics affected the impact of silent spread
We then focused on the spread of Alpha in six countries where national genomic investigations estimated the incidence of the Alpha variant in early January 2021: Denmark, France, Germany, Portugal, Switzerland, and the USA. We used a stochastic model (autochthonous model A)[40] to simulate chains of transmission generated by infections introduced from the UK as predicted by the international dissemination model described above. The model used

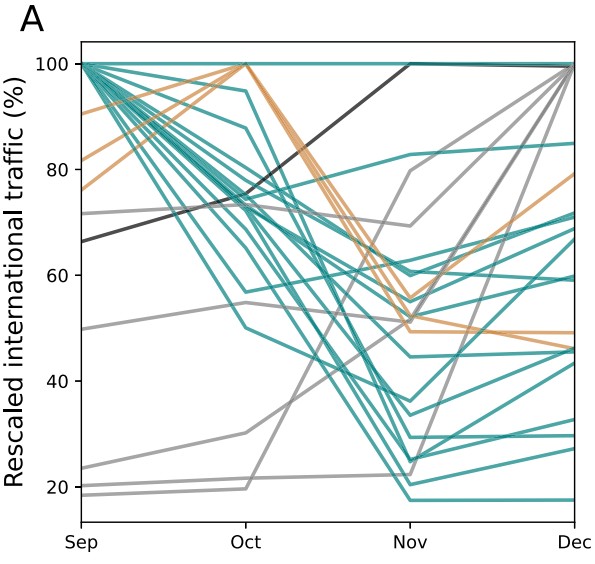

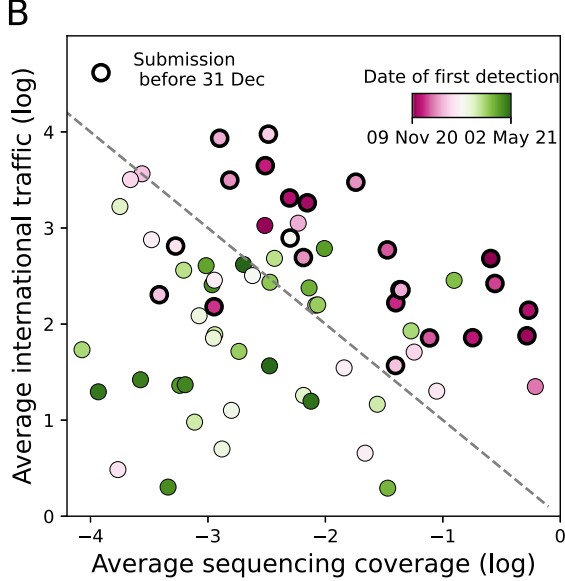

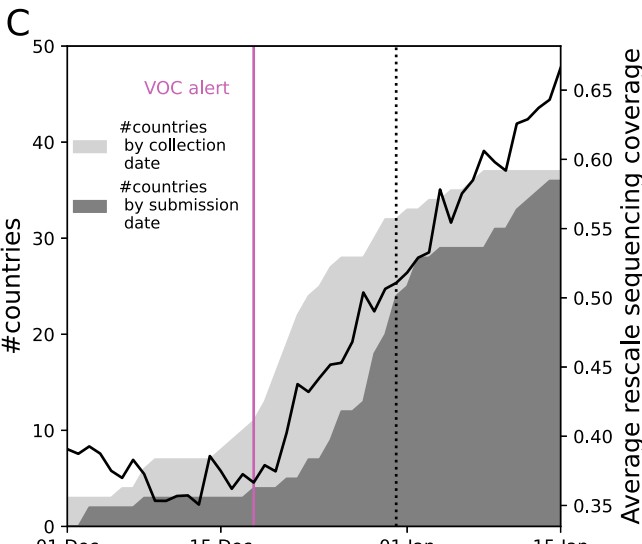

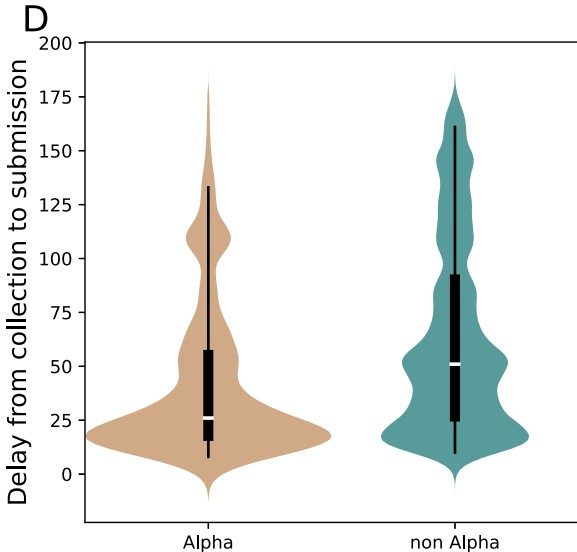

**Fig. 1 | Factors associated with the pattern of observed Alpha dissemination.**
**A** Change in outbound international traffic from the UK over time, including air-travel, train, ferry and Channel Tunnel[59]. The 73 countries contributing to GISAID during 1 Sep 2020–31 Dec 2020 are shown as an example. Traffic is rescaled to the maximum over the period. To improve readability, different months of traffic maximum are associated with a different color. **B** Date of first detection, i.e. collection of the first Alpha sequence submitted to GISAID, for each of the 73 countries as in (**A**), according to sequencing coverage and international traffic (passengers/day) averaged over 1 Sep 2020–31 Dec 2020. For each day, the sequencing coverage of a country is defined as the number of collected SARS-CoV-2 sequences on GISAID −regardless the date of submission−divided by reported cases. The dashed line provides a guide to the eye, as, under simplifying assumptions[44,81], we expect the date of first detection to be a function of log(sequencing coverage) + log(traveling flaw) (Supplementary Information). **C** Number of countries with at least one Alpha submission plotted by date of collection and date of submission. The black line shows the average rescaled sequencing coverage. In each country, the sequencing coverage was rescaled by the maximum over the period displayed in the plot to highlight the trend. Countries' rescaled time series were then averaged. For the sake of visualization, the sequencing coverage is here smoothed over a 2 weeks sliding window. The purple line indicates the date of Alpha international alert (18 Dec 2020). The dashed black line indicates the censoring date used in the analysis (31 Dec 2020). **D** Distributions of delay (in days) from collection to submission for Alpha and non-Alpha sequences collected outside the UK from December 2020 to mid-January 2021 and submitted up to June 2021 (non-Alpha sequences $n = 149699$, Alpha sequences $n = 6992$). Boxplots represent the median (white bar), the quartiles and the 95% range (whiskers). The violin plot shows the Kernel estimation of the underlying distribution. Additional details are reported in Supplementary Fig. 1.

country-specific time-varying reproduction number, overdispersion in transmission, and a 60% transmission advantage of Alpha over the wildtype[28,29,41]. The model reproduced the same trend of the observed Alpha cases with a case ascertainment fraction around 50% (Fig. 4A). Incidence in the USA was underestimated, possibly due to heterogeneity in the different states. To test a finer spatial resolution

we retrieved Alpha frequency data for California, Florida, and New York City, obtaining a good match with the data for California and New York City and an under-estimation (within the range of possible stochastic outcomes) for Florida (Supplementary Fig. 4). To test the robustness of these predictions, we used a second model with age-structure, temporal variation in social contacts due to restrictions,

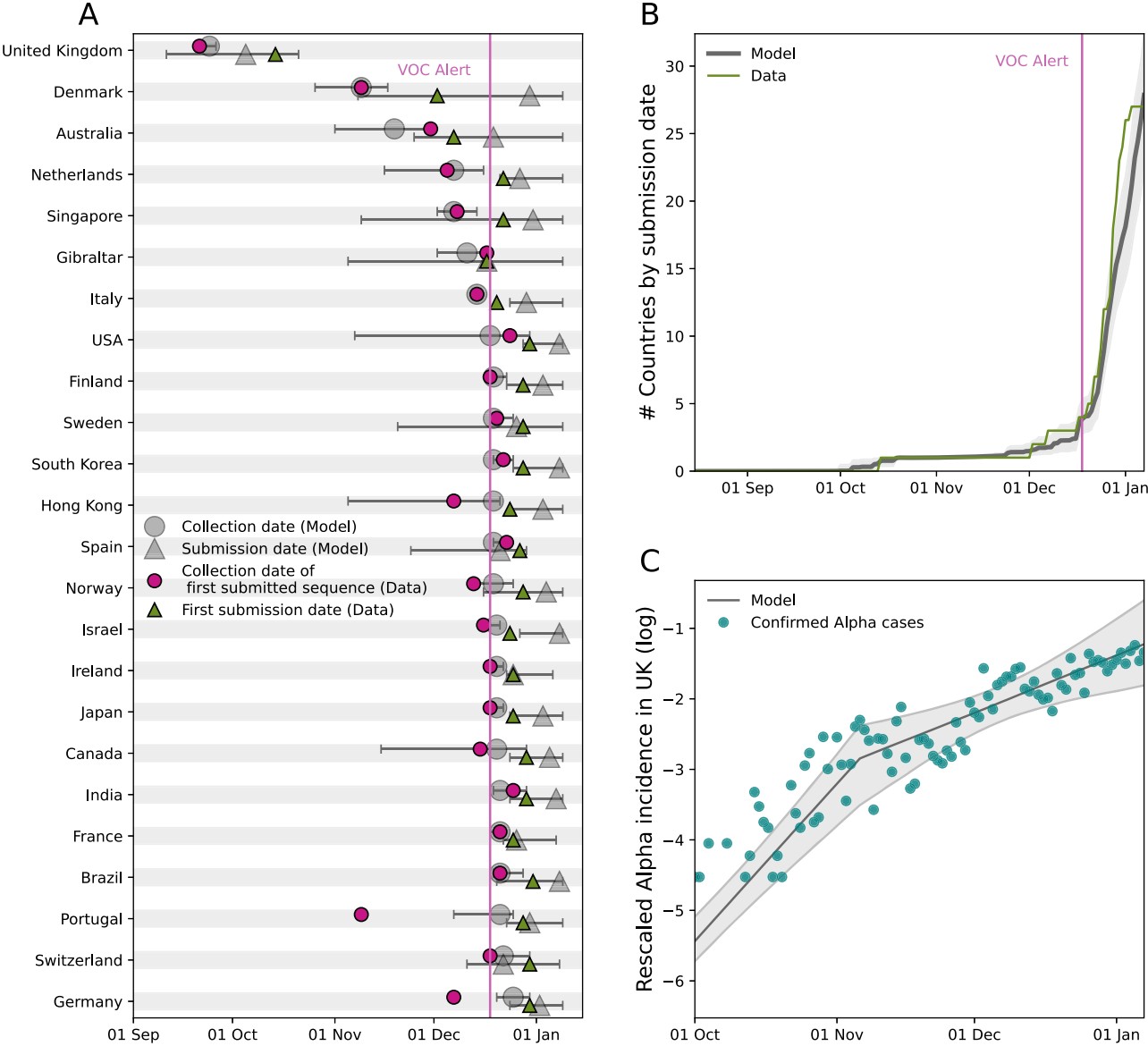

**Fig. 2 | Comparison between the international dissemination model and the data. A** Date of collection of the first Alpha sample submitted to GISAID and corresponding date of submission for the 24 countries submitting Alpha sequences before 31 Dec 2020. Data are shown by purple circles (collection) and green triangles (submission). Median date obtained from the model is indicated by gray circles (collection) and gray triangles (submission). The horizontal bars display the 95% prediction interval over n = 500 simulations. **B** Median model predicted cumulative number of countries submitting a first Alpha sequence to GISAID compared with observations. In panels A and B, the purple vertical line indicates the date of Alpha international alert (18 Dec 2020). **C** Alpha incidence in the UK[36] and median model-predicted epidemic profile in the UK. Both model predictions and data are rescaled to the sum over the period considered to allow comparing the profiles of the curves. To account for testing delays model predictions are shifted right of one week. The gray colored ribbon represents the 95% credible interval.

and the co-circulation between Alpha and wildtype that was calibrated and validated for France[42,43] (autochthonous model B) finding also in this case a good agreement (inset of Fig. 4A).

Besides supporting our estimates of Alpha dissemination out of the UK, the reconstruction of local epidemics outside the UK allowed investigating the potential impact of silent spread in the six focal countries. The estimated Alpha cases as of 31 Dec 2020 broadly scaled with the international traffic connecting the country with the UK, showing the important role of importations in determining local Alpha epidemic size (Fig. 4B). Still, potential consequences of silent spread could only be gauged by taking into account changes in local transmission (Fig. 4C). For example, while the first detected case in Germany and Switzerland had been collected with a similar delay from the predicted date of first importation, the reproductive ratio $R_t$ in Germany had generally been larger than in Switzerland during the period.

Therefore, the seeding of transmission chains still active at the end of the year in Germany could take place well before the first detected case was collected for the first time in the territory, while in Switzerland ~50% of the transmission chains started after first virus detection (Fig. 4C). Overall, later seeding of active chains was associated with smaller average $R_t$ over the period (Fig. 4D), but not with the reduction in traveling (Supplementary Fig. 5). Therefore, our analysis suggests that low levels of local $R_t$ enhanced the relative contribution of late importations, potentially countering the consequences of late detection.

## Discussion

Genomic surveillance has been a major advancement in monitoring the spread of SARS-CoV-2 after initial emergence. However, interpreting these data is complicated as they do not follow a pre-

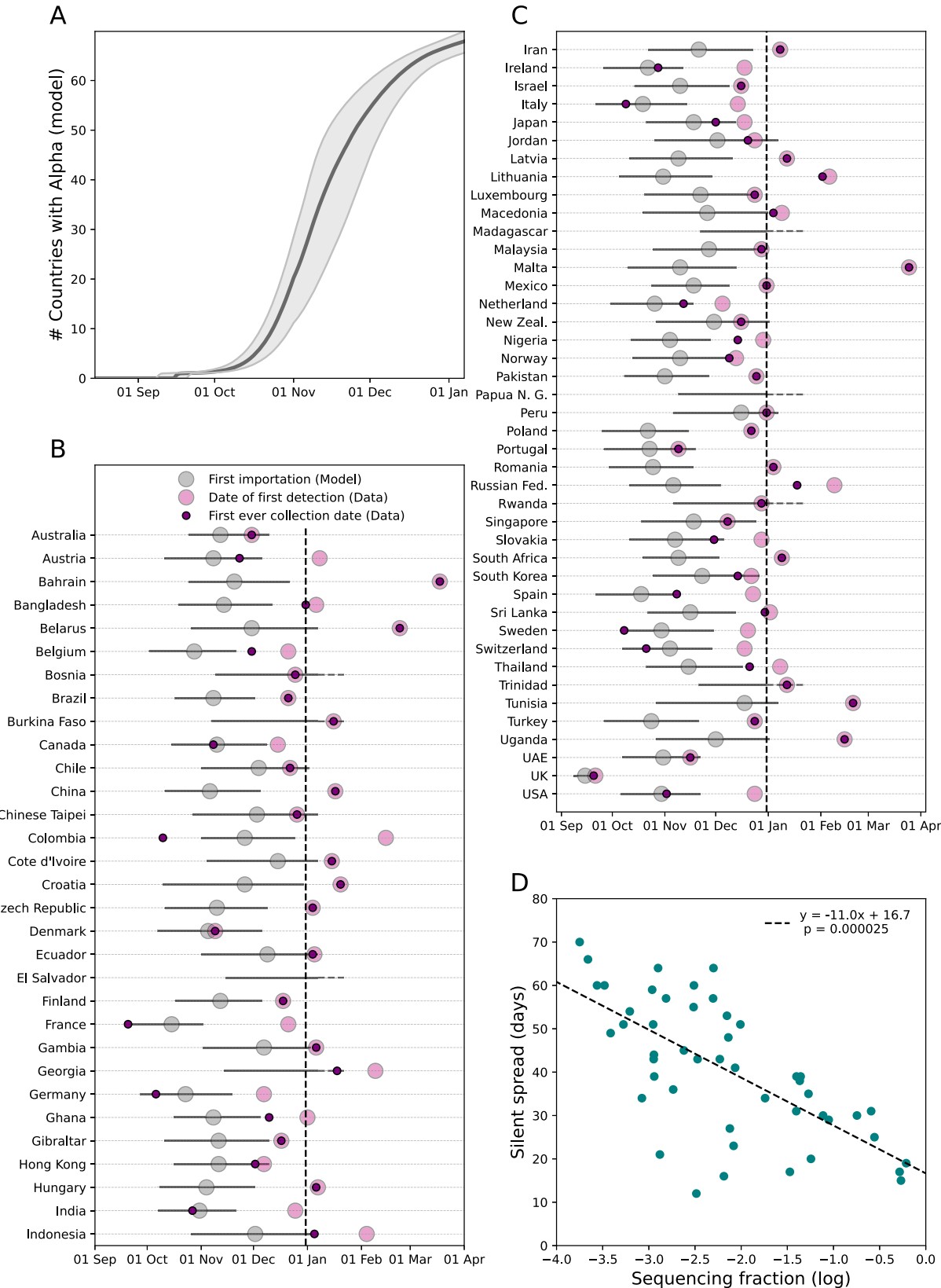

established and coordinated sampling design. Retrospective analyses of the past dissemination of VOCs can provide epidemiological knowledge that enables us to better respond to future viral emergence events. Here, taking the initial Alpha spread as an example, we showed that several components of the highly heterogeneous epidemic context had to be taken into account for interpretation.

Previous studies focused on traveling flows to explain the arrival of a first infection into a new country[44–48]. Yet differences in genomic surveillance capacity, over four orders of magnitude across countries during Alpha emergence, profoundly affected the introduction-to-detection delay with a logarithmic decrease in sequencing coverage. Furthermore, extraneous events like the international alert further

**Fig. 3 | Timing of first importation and silent spread as estimated by the international dissemination model. A** Cumulative number of countries with an Alpha introduction as predicted by the model. The quantity is computed from the median predicted date of introduction in each country with 95% prediction interval obtained over n = 500 simulations. **B, C** Median date of first introduction when occurring before Dec 31 (vertical dashed line) for each country estimated by the model with 95% prediction interval computed over *n* = 500 simulations. For each country, we report the date of first Alpha detection (i.e. collection of first submitted sequence) (light pink) and the date of the first ever collected Alpha sequence (dark pink) from the data. For El Salvador, Papua New Guinea and Madagascar, no Alpha sequence had been reported before June 2021. **D** Duration of silent spread in days vs sequencing coverage. The distribution of the durations of silent spread is reported in Supplementary Fig. 3. Duration of silent spread is computed as the difference between the median date of first detection and the median date of first introduction as predicted by the model. We restricted the analysis to countries for which both first introduction and first detection were predicted to occur before 7 Jan 2021. Dashed line represents least-squares linear regression. *P*-value is computed from Wald test with t-distribution.

altered the speed of variant detection. These strong spatiotemporal changes in genomic surveillance partially masked the true pattern of Alpha invasion, to the point that the correlation between the dates of detection and the international traffic was poor in the first 24 countries reporting Alpha (spearman correlation 0.24, *p* = 0.3). Yet the good fit obtained with the international dissemination model and the increase of Alpha epidemic size with traveling flow (Fig. 4B) both suggest that traveling flows were a driver of viral spread, in agreement with other works[19,44–47,49]. A more uniform sequencing collection protocol would have provided a coherent view of Alpha propagation improving public health awareness and response. This highlights the importance of eliminating surveillance blind spots by increasing sequencing in countries with poor surveillance[23].

According to our model, Alpha was introduced in more than 60 countries before the international alert. This provides evidence that when an emerging pathogen is not reported in a given destination country, it may likely be due to the surveillance system not yet being able to detect it. The alert triggered heightened genomic surveillance worldwide, reinstated lockdown measures in the UK, and resulted in border screening and travel bans in countries connected to the UK[23,28–30]. However, international response arrived at a moment in which Alpha was already widespread in several countries, preventing containment. Improving surveillance across countries would reduce the time from importation to detection, but it would still clash with the delay needed to recognize a novel variant as a VOC. A lineage with important mutations can be identified relatively quickly if sequencing coverage is high enough[23,24,27], although the assessment of clinical risk is slower[24]. Lineages have shown the ability to become dominant without any increase in fitness in particular epidemiological contexts[50], while others like Beta remained at low frequency despite mutations of clinical importance. A more rapid recognition of Alpha as a VOC could have advanced the response by health authorities to delay the establishment of Alpha during a time when vaccination became available in some countries[21]. Similar delays in declaring a VOC were also observed for subsequent VOC episodes[19]. This underlines the complexity of the interpretation of a context with emerging new variants[51]—especially when major known drivers such as international travel are in place—and of the decision-making for public health response.

The growing Alpha epidemic in the UK allowed dissemination despite the drop in international traffic out of the UK and the social restrictions in many countries. For instance, while UK travelers to France dropped by 56% in November compared to September, the number of Alpha-infected travelers to France still grew from 1 to 10 daily over November 2020 according to our model. The lockdown implemented in France at this time likely did not prevent local transmission because Alpha was more transmissible. Local restrictions may however delay successful invasion, as was apparent from the in-depth analysis of the six destination countries: a lower local reproductive ratio delayed the seeding of local transmission chains following importations up to one month. Although with the same analysis we could not address the consequences of the decline in travel, we expect that when local transmission is limited by control measures, introductions from the country of origin contribute more substantially to the epidemic at destination[20]. We can thus hypothesize that limiting importation early could act synergistically with local restrictions to limit the size of the VOC epidemic. Still, we expect that the fine tuning between different factors (e.g. quality and extent of borders control and timing of their implementation[22,52,53]) can affect the impact of travel restrictions.

Following Alpha, other SARS-CoV-2 variants raised concern due to their rapid emergence and spread, namely Beta, Gamma, Delta, Omicron and its sublineages. Undetected introductions and silent spread were likely common to all variants, although the epidemic context progressively changed between 2021 and 2022. The rise in international mobility and social contacts accelerated the spread of Delta and Omicron[19]. This has reduced the window for public health response requiring an intensification of virus genomic surveillance to enable authorities to identify variants in time. However the high costs of genomic surveillance and the phasing out of the pandemic have now reduced our ability to detect future VOC emergence events. The Alpha experience shows the importance of designing sequencing protocols able to balance sustainability and detection capacity by meeting the minimal requirements of sequencing extent and reporting delay—e.g. sequencing 0.5% of cases with a turnaround time smaller than 21 days as previously proposed[23], and by leveraging information from multiple sources, including wastewater and animal surveillance[54,55]. Importantly, this study also highlights that the knowledge of surveillance extent and protocol adopted by countries is key to real-time data analysis to better assist risk assessment and intervention planning. This would be facilitated by the widespread adoption of pre-established surveillance protocols.

Our study is affected by a number of limitations. First, sequencing coverage was computed at the country level and no distinction could be made for traveler vs. local cases due to the poor available information on testing rate among travelers[18]. We dealt with this by allowing an increase in detection after the Alpha alert. Second, we analyzed here the period before 31 Dec 2020. This time window was long enough to cover the seeding from the UK to the destination countries and observe the consequent onset of local transmission. At the same time, the window is sufficiently short to assume in first approximation the UK to be the source of Alpha spread, before large epidemics in other countries became the dominant source of traveling cases. Extending the analysis to a longer period would require a more general framework that can be the subject of future work. Third, it is not possible to set a cut-off between real-time and retrospective surveillance when computing sequencing coverage from GISAID metadata. The computation of sequencing coverage being affected by retrospective surveillance could potentially overestimate the extent of the real-time genomic surveillance. Fourth, we have here defined the date of first Alpha detection in a country as the date of collection of the first sequence submitted to GISAID. Reporting of variants of interest to local public health authorities can be indeed more rapid than submitting the sequence to GISAID. Still, we acknowledge that this may depend on the country and stage of the invasion, e.g. before and after the alert. In addition, the public sharing of a variant's sequence enables the recognition of its presence in a given territory by a larger public, including health authorities and the scientific community worldwide.

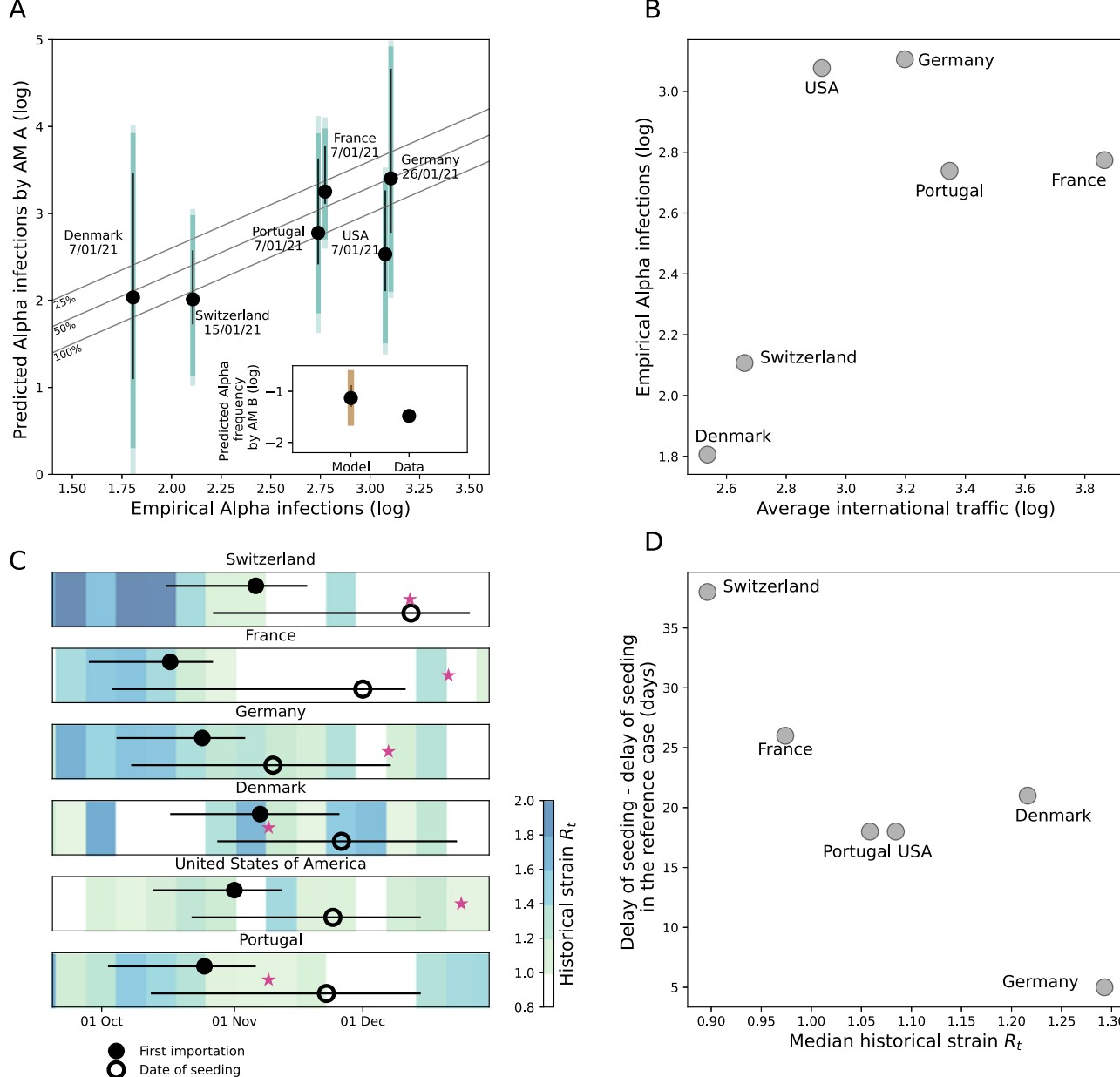

**Fig. 4 | Local spread of Alpha in six destination countries. A** Model vs. empirical Alpha infections. In the main plot, the empirical estimates of Alpha cases are computed by multiplying the Alpha frequency from virological investigations by the reported COVID-19 incidence at the same date—the date is indicated in the plot. Model estimates are obtained with the autochthonous model A (AM A in the plot). Gray lines show ratios of 100%, 50% and 25% between observed and predicted infections attributable to reporting. In the inset, the frequency of Alpha in France obtained from the autochthonous model B (AM B in the plot) is compared with the empirical data. In both panels, black error bars indicate the prediction interval over 500 stochastic simulations obtained with the median volume of Alpha introduction, output of the international dissemination model assuming a 7-day delay between case and infection. Dark colored bars account for the variability in the output of the autochthonous models accounting for the upper and the lower limit of the prediction interval of the Alpha introductions as given by the international dissemination model. Light colored bars account for variability in the delay from infection to case reporting (ranging from 4 days to 10 days). **B** Empirical Alpha infections vs average international traffic. **C** Comparison between the date of first introduction as predicted by the international dissemination model and the seeding time of the transmission chains survived until 31 Dec 2020, predicted with the autochthonous model A. Circles indicate medians and segment the 95% prediction interval. Colors indicate the effective reproduction number of the historical strain, $R_t$, computed from weekly mortality data (Methods). The star shows the date of first Alpha detection as a comparison. **D** Difference between the median delay of seeding predicted by the autochthonous model A and the same quantity in the reference case—i.e. when $R_t$ is the same in all countries and traveling fluxes do not change in time, plotted against the median $R_t$ during the period from first introduction to seeding.

To conclude, by jointly modeling epidemic dissemination and observation based on GISAID submissions we have quantified Alpha silent spread in countries outside the UK unveiling its link with international travel and sequencing coverage. Our results show that the duration of Alpha silent spread varied from days to months. Strong spatiotemporal heterogeneities in surveillance provided a major obstacle to data interpretation. Still, restrictions in place in destination countries may have delayed the establishment of local transmission and partially mitigated the negative consequences of late detection and response. By the time a new variant is recognized as a potential

threat, surveillance authorities of countries outside the variant source should be prepared for the variant being potentially already present in the territory. Enhancement in local screening and measures aiming at containing local transmission are thus key ingredients of a response plan. Taken together these findings provide lessons learnt for the future management of SARS-CoV-2 variants. Beyond that, retrospective reconstructions of SARS-CoV-2 spread are essential to improve computational modeling and public health knowledge to better guide the response to future spread of viruses with zoonotic and pandemic potential.

## Methods
### Data
**GISAID records.** While we did not use any actual sequences in this study, from GISAID entries[3] we retrieved collection dates, submission dates, information on lineage (i.e. whether it was Alpha or not) and country for all human SARS-CoV2 sequences submitted between 15 Aug 2020 and 1 Jun 2021 included (n = 1,735,675 downloaded on 2 Jun 2021). Data in GISAID originated from 144 countries, however, only 73 countries had submitted sequences collected between September 2020 and December 2020. We used GISAID entries to determine the date of the first submission of an Alpha sample in each country and the respective date of collection, the latter defined as the "detection date". Assuming that detection occurs at the time of sample collection corresponds to the optimistic hypothesis that surveillance authorities are informed right after a sample is collected. We also computed the date of the first collection ever of an Alpha sample in each country, irrespective of the date submitted. Finally, we determined the distribution of delays from collection to submission and the sequencing coverage from the number of sequences by country and date of collection (see below). For GISAID sequences missing a collection date (3%), we imputed the missing date with a date selected at random from the sequences with complete data submitted in the same week and country. We resorted to imputation instead of inferring a statistical model because of the small percentage of missing records. In addition, the strong spatiotemporal variations displayed by the data could be hard to capture by a statistical model.

**COVID-19 cases and death data.** We retrieved the daily number of COVID-19 cases by country from the COVID-19 data repository hosted by the Center for Systems Science and Engineering at Johns Hopkins University (CSSE)[56] to compute the sequencing coverage. Incidence of Alpha cases in the UK was obtained from the "Variants of Concern: technical briefing 7−Data England" report[36]. We used the weekly deaths time series from the European Center for Disease Prevention and Control[57] (downloaded on 1 Jul 2021) to compute the time varying reproduction ratio in Denmark, France, Germany, Portugal, Switzerland and the USA.

**Travel data.** Travel flow from the UK to destination countries was reconstructed combining air travel data, estimates of passengers via train, Channel Tunnel and ferries. We computed probabilities of travel assuming a catchment population of 36 M for London airports. More precisely:

- Air travel data were obtained from the International Air Transport Association (IATA)[58]. It comprised the monthly number of passengers outbound from English airports by country of destination. From the monthly data we computed an averaged daily flux of passengers over the month. For each country, we aggregated all passengers directed to the country and leaving from all airports of London.
- Eurostar rail passenger numbers going each day to France, Belgium and the Netherlands were estimated as in[59], assuming a 95% reduction due to the COVID-19 pandemic[60].

- We used the monthly number of cars crossing the Channel Tunnel[59,61] to derive an averaged daily flux of passengers over the month. We assumed that 1.5 passengers travel on average for each car[59] and that the repartition of passengers among countries in continental Europe is the same as for trains.
- Numbers of passengers via ferries to France, Belgium, the Netherlands, Spain and Ireland were obtained by ref. 62. We used monthly data to compute an averaged daily flux of passengers over the month.

**Virological investigation records.** National investigations were conducted in a number of countries in early January. Through bibliographic search and via social media we gathered the data from virological surveys or extensive screening for Denmark, France, Germany, Portugal, Switzerland, the USA. These surveys give an estimated frequency of Alpha infections for the cases detected a given day (or a given time period). We also considered the daily number of detected cases on the day of the survey (or the midpoint of the time period) from CSSE. From these two numbers, we calculated the number of detected Alpha cases. In Supplementary Table 1 we report the source, the date of the survey, detected Alpha frequency, and the number of Alpha cases computed for each country. In Supplementary Fig. 4, we also analyzed three locations in the US, i.e. Florida, California and New York City. Sources for these data are reported in Supplementary Table 2.

### Data processing
**Sequencing coverage.** The sequencing coverage was computed for each day and each country as the number of sequences collected after imputation divided by the number of cases. In Fig. 1C, we smoothed the sequencing coverage with a two-week sliding window to highlight the general trend.

**Delays from collection to submission.** We computed the collection-to-submission times in different ways before and after the Alpha alert on 18 Dec 2020. Before the alert, we hypothesized that Alpha sequences would be reported with the same time pattern as other sequences. We therefore computed a delay distribution by country and by date of collection using all GISAID sequences as $\pi_c(d;u) = \frac{n_{u+d,c}}{N_{u,c}}$ where $d$ is the delay, $N_{u,c}$ the number of sequences collected on day $u$ in country $c$ and $n_{u+d,c}$ those submitted on date $u+d$. For sequences collected after the alert of 18 Dec 2020, we accounted for the different delay distribution for Alpha and other sequences. Due to the limited number of Alpha sequences collected outside the UK soon after the alert we aggregated all data collected outside the UK, thus defining an average Alpha delay distribution for all countries. We then used a 3-day smoothing time window, where length 3 was chosen as the best compromise to smooth out fluctuations without masking meaningful trends. We therefore computed $\pi_c(d;u) = \frac{n_{d+u}}{N_u}$ with $N_u$ the number of Alpha sequences collected between day $u-1$ and $u+1$, and $n_{u+d}$ the number of those sequences submitted after $d$ days for each country $c$. Delays from collection to submission are reported in Fig. 1D and Supplementary Fig. 1. In the sensitivity analysis we computed the Alpha collection-to-submission delays after 18 Dec 2020 separately for each country. We used a longer smoothing time window (7 days instead of 3 days) to compensate for the geographic disaggregation.

### International dissemination model
We model the observed data consisting in date pairs $\{S_c, T_c\}$ by country, where $S_c$ is the date of first submission of an Alpha sequence to GISAID and $T_c$ the corresponding date of collection in country $c$. The model is based on the following assumptions: i) Alpha incidence in the UK grows exponentially with a piecewise exponential rate to account for the autumn lockdown; ii) imported cases are proportional to international traffic; iii) collection and sequencing of a sample from an

imported case of SARS-CoV-2 Alpha and its submission on GISAID is proportional to sequencing coverage and the detection-to-submission delay computed from GISAID metadata.

More precisely, we first described incident Alpha infections in the UK at time $t$ as exponentially growing with time according to $inc_{UK}(t) = \exp(\sum_{T_0}^{t} r(u))$, where $T_O$ is fixed at 15 Aug 2020[33], the date when the risk of emergence starts and $r(t)$ the daily exponential growth rate. The daily exponential growth rate in the UK was considered piecewise constant, $r_1$ up to Nov 5th, 2020, when the UK entered a lockdown, and $r_2$ afterwards. In other words, $inc_{UK}(t)$ was a "two-slope" exponential, growing as $\exp(r_1 t)$ before Nov 5 and as $\exp(r_2 t)$ afterwards. We also explored a model with no change of slope and two changes of slopes (at 5 Nov 2020 and at 2 Dec 2020, beginning and end of the lockdown respectively) in the sensitivity analysis.

In the UK, the number of Alpha sequences collected depended on incidence and sequencing coverage as

$$\lambda_{UK}^{*}(t) = K_{UK} s_{UK}(t) \sum_{j=0}^{J} inc_{UK}(t-j), \qquad (1)$$

where $s_{UK}(t)$ is the sequencing coverage on day $t$, $J$ the duration of incubation and $K_{UK}$ the detection probability. For the incubation period we used 5 days[63] and tested 4 and 6 days in the sensitivity analysis. We considered that one case out of 4 would be tested ($K_{UK} = 0.25$)[64].

Consistently with[4,32] we assumed that the first case reported to GISAID in each country outside the UK was an imported case, infected in the UK but discovered abroad. Thus, we modeled detection and sequencing in countries outside the UK without the need to model local variant growth. There, the expected number of sequences collected at time $t$ in country $c$ additionally accounted for traveling as

$$\lambda_{c}^{*}(t) = K_c p_c(t)/N s_c(t) \sum_{j=0}^{J} inc_{UK}(t-j), \qquad (2)$$

where $p_c(t)/N$ is the fraction of the population traveling from the catchment area of the London airports to country $c$ on day $t$ with $N = 36$ millions inhabitants the population of the area, and $s_c(t)$ the sequencing coverage in country $c$ on day $t$ and $K_c$ the fraction of imported infections being detected as COVID-19 cases. We assumed detection of imported cases to be higher than the detection of local cases, thus we used $K_c = 0.5$ ($>K_{UK}$). In the sensitivity analysis, we tested all airports of England, instead of airports of London, as the origin of Alpha infected travelers, and $K_c = 0.25$. Finally, we allowed for an increase in collection of Alpha sequences among travelers relative to others after the alert of 18 Dec 2020 due to increasing sampling of travelers from the UK[4,32] using a multiplicative factor $\gamma$. Therefore, the expected number of collected Alpha sequences on day $t$ is $\lambda_c(t) = \lambda_{c}^{*}(t)$ before 18 Dec 2020 and $\lambda_c(t) = \gamma \lambda_{c}^{*}(t)$ afterwards. Taking into account collection-to-submission time, the expected number of sequences submitted at time $t$ in country $c$ is therefore $\alpha_c(t) = \sum_{u \le t} \lambda_c(u) \pi_c(t-u,u)$, and the probability that a sequence submitted on day $t$ was collected on day $u$, with $u \le t$, is $\lambda_c(u) \pi_c(t-u,u)/\alpha_c(t)$.

To write up the likelihood of observations, we considered that the model described the dynamics of collection and submission until the end of 2020. We assumed Poisson variability in the number of Alpha infections and computed the probability that an Alpha sequence is submitted on GISAID for the first time on date $S_c$ in country $c$ as

$$P(S_c) = \exp\left(-\sum_{u < S_c} \alpha_c(u)\right)\left(1 - \exp(-\alpha_c(S_c))\right) \qquad (3)$$

The log-likelihood of the data in the model was:

$$\log L(\{r_0, r_1\}, \gamma; \{S_c, T_c\}) =$$

$$= \sum_{c:S_c \le D} \log(1 - \exp(-\alpha_c(S_c))) + \log(\lambda_c(T_c)\pi_c(S_c - T_c, T_c)/\alpha_c(S_c))$$

$$- \sum_c \sum_{T_0}^{S_c} \alpha_c(u), \qquad (4)$$

where the first sum runs on countries where an Alpha sequences was submitted before date $D$ (= 31/12/2020) and the second runs in all countries. The summary of all fixed parameters and their values is reported in Supplementary Table 3.

The model likelihood was explored with a Metropolis-Hastings procedure using R v4.3. We used an Exp(0.1) exponential prior on the first exponential growth rate $r_1$, a N(0,1) prior on second growth rate $r_2$ to allow for negative growth and an Exp(0.01) prior for the increase in sampling $\gamma$ (Supplementary Table 4). Unless stated otherwise, 3 chains were run in parallel for 100000 iterations, with the first 50000 discarded as burn-in, the second half was thinned (1 iteration every 25) for a final posterior sample of size 2000. Convergence of the chains was checked visually (Supplementary Fig. 2). Estimates and credible intervals for the fitted parameters are reported in Supplementary Table 5 (baseline values, first row).

We computed the predictive distribution for the date of detection given the actual travel and sequencing coverage as

$$F_c(t;p_c,s_c,K_c) = 1 - \exp\left(-\int_{T_0}^{t} \lambda_c(u;p_c,s_c)du\right) \qquad (5)$$

using the posterior sample and computed 95% prediction intervals from these samples.

We finally computed the model-predicted date of first introduction in country $c$ as the distribution $F_c(t;p_c,1,1)$ in each country, assuming that 100% sequencing occurred ($s = 1$) and all cases were detected ($K = 1$).

We computed predictive distributions from the model using parameters taken in the posterior distribution as follows (where the "hat" notation corresponds to the estimated value):

- Expected incidence in the UK:

$$inc_{UK}(t) = \exp\left(\sum_{T_0}^{t} \hat{r}(u)\right) \qquad (6)$$

- Distribution of time of emergence in the UK:

$$P(T_e < t|T_e < T_{UK}) = 1 - \exp\left(-\sum_{T_0}^{t} \hat{r}(u)\right)\bigg/\left(1 - \exp\left(-\sum_{T_0}^{T_{UK}} \hat{r}(u)\right)\right) \qquad (7)$$

- Cumulated distribution of date of first submission:

$$P(S_c \le t) = 1 - \exp\left(-\sum_{u \le t} \hat{\alpha}_c(u)\right) \qquad (8)$$

- Cumulated distribution of date of first introduction:

$$P(I_c \le t) = 1 - \exp\left(-\sum_{u \le t} \hat{\lambda}_c^1(u)\right) \qquad (9)$$

with

$$\lambda_c^1(t) = p_c(t)/N \sum_{j=0}^{J} inc_{UK}(t-j) \qquad (10)$$

the number of (detected and undetected) infections.

To visualize goodness of fit, we computed the cumulated number of countries submitting an Alpha sequence by date $t$ as $\sum_c P(S_c \le t)$, and for the countries reporting an Alpha sequence, the cumulative distribution of introduction date conditional on submission date, $P(I_c \le t | S_c)$.

## Autochthonous model A

To simulate the number of Alpha variant infections at the beginning of 2021 in each country of interest, we used the daily rates of importation as estimated from the international dissemination model $\lambda_c^1(t))$ and simulated the subsequent stochastic outcome of each imported infectious individual in the destination country. The different Alpha epidemic clusters initiated by each importation were assumed to be independent. The stochastic epidemic growth model has been described elsewhere[40]. For each day since $T_0$ and each country of destination, we drew the number of imported infections in a Poisson distribution with rate $\lambda^1_c(t)$. Then, starting with each imported infection, we simulated an epidemic chain assuming that each infected individual produced a number of secondary infections according to a negative binomial distribution with mean $(1+\alpha)R_t$ and dispersion parameter $\kappa = 0.4$, where $R_t$ is the effective reproduction number at date $t$ and $\alpha = 0.6$ is the transmission advantage of the Alpha variant relative to the historical strain, assumed to be the same in every country[41]. The generation time distribution was gamma with mean 6.5 days and s.d. 4 days (shape 2.64, scale 2.46)[29]. To compute the effective reproduction number $R_t$ of the historical strain from mortality data, we computed first the daily exponential growth rate as $r_t = 1/7 \log(D_{w+1}/D_w)$ where $D_w$ is the number of deaths in week $w$. To account for the lag between disease onset and death (approx. 3 weeks), we considered that this exponential growth rate applied to infections for days $t$ in week $w-3$. We finally computed $R_t = \int_0^\infty \exp(-r_t\tau)g(\tau)d\tau$ with $g(\tau)$ the generation interval distribution[65]. Note that the calculation of $R_t$ in this way is robust to under reporting biases, provided that the reporting ratio does not change substantially over the period. This approach yielded estimates similar to the Epiestim method[66].

The model was implemented in C + + (v11). In the simulations of epidemic clusters, the code loops over time, starting from one infected individual at the day of importation, and ending at 31 Jan 2021. Time was discretized in time-steps of 0.1 day. The secondary infections are added to their (future) date in the incidence table, and the code proceeds to the next infected individual at this time step, then to the next time-step. Five hundreds (500) replicate simulations were obtained for each country to account for stochastic variability in the number and timing of importations and growth of local epidemics.

Number of infections output of the model were compared to the empirical number cases estimated from the virological survey. Assuming a delay between infection and case detection of one week, empirical cases were compared with model-predicted Alpha infections 7 days before. Since delay in reporting may vary from one country to another—some countries report cases by date of testing, others by date of notification, data may be smoothed, etc.—we also tested delays of 4 and 10 days.

## Autochthonous model B

We used a stochastic discrete age-stratified, two-strain transmission model to simulate the epidemic dynamics in France generated by the estimated Alpha importations[42,43,67].

The model integrates data on demography, age profile, social contacts, mobility and adoption of preventive measures. Four age classes are considered: [0–11], [11–19], [19–65] and 65+ years old (children, adolescents, adults and seniors respectively). Transmission dynamics follows a compartmental scheme specific for COVID-19 where individuals are divided into susceptible, exposed, infectious, hospitalized and recovered. The infectious class is further divided into prodromal, asymptomatic and symptomatic. Susceptibility and transmissibility depend on age[68-70]. Transmissibility also depends on the level of symptoms[71-74].

Contact matrices are setting-specific. Contacts at school are modeled according to the French school calendar, while those at work and on transports according to the workplace presence estimated by Google data[75]. During the different stages of the pandemic, physical contacts are modulated based on surveys on the adoption of physical distancing[76], self-protection[42], and assuming a reduction in contacts due to severe symptoms. The integration of all these data allows for capturing the social distancing restrictions put in place in France to curb the second wave, namely a lockdown with schools open[77] from week 44 (starting October 31, 2020) to week 51 (ending December 15, 2020).

The model was previously used to respond to the COVID-19 pandemic in France in 2020[42,43,63,78], assessing the impact of lockdown[63], of night curfew[43] and of the reopening of schools[78], estimating the underdetection of cases[42], and anticipating the impact of the Alpha variant in France[43]. In particular, we used, here, the same two-strain version of the model developed to study the impact of January 2021 curfew in France on the Alpha circulation in the territory[43], with same parametrization and same transmissibility calibrated to national daily hospital admission data[79]. This accounts for the co-circulation of Alpha variant and the historical strains, and assumes complete cross-immunity between the two strains, higher hospitalization rate and an increase in transmissibility of 50% for Alpha[28]. We also tested a 60% advantage in transmission, finding that results were robust. Values of other key parameters are generation time equal to 6.6 days, and incubation period 5.2 days. Other parameter values are reported in ref. 63. The model was implemented in Python 3.8.5.

We simulate the epidemic dynamics using the output of the international dissemination model as seeding for the dynamics. At each date, we extract the number of prodromal adults infected with the variant from a Poisson distribution with mean equal to the traveling cases at that date obtained from the international dissemination model. We repeat this extraction for each of the 500 stochastic runs performed and we simulate the resulting outbreak. We then compute the proportion of Alpha on January 8 and compare it with the proportion identified by the first large-scale genome sequencing initiative (called Flash #1)[41] conducted in the country on January 7-8, 2021 (Alpha proportion in France equal to 3.3%).

## Seeding time of active transmission chains

The time of seeding of a transmission chain still active at a reference end time (time $T_R$) is uniformly distributed over the range of possible introduction times when the exponential growth rate $r$ is the same in the place of origin (here the UK) and in the destination country and traveling flows are constant over time. This is because starting from the date of emergence $T_E$, the number of introductions in the destination country at some time $t_I$ will be proportional to $\exp[r(t_I - T_E)]$ and each case introduced will cause $\exp[r(T_R - t_I)]$ cases at time $T_R$, so that the overall number of cases at time $T_R$ is $\exp[r(t_I - T_E)] \exp[r(T_R - t_I)] = \exp[r(T_R - T_E)]$ irrespective of the actual date of introduction. Therefore, date $(T_E + T_R)/2$ is the expected median introduction date in this simple scenario of constant exponential growth rate and traveling.

We therefore used the autochthonous model A to reconstruct the distribution of the seeding times for the transmission chains still active

on December 31st, 2020. We computed the distribution of seeding times and the difference between the median of this distribution and the expected median under the constant exponential growth rate and traveling described above. The extent of this difference illustrates the effect of the actual change in epidemic growth rate and traveling flows on seeding success. We are here interested on how this quantity changed across the six countries. We found that it increased for lowering values of $R_t$. This show that low values of $R_t$ were likely hindering the seeding of local transmission chains by the introduced cases, making the late importations comparatively more important.

### Reporting summary

Further information on research design is available in the Nature Portfolio Reporting Summary linked to this article.

## Data availability

The findings of this study are based on metadata associated with a total of 1,735,675 sequences available on GISAID and submitted between 15 Aug 2020 and 1 Jun 2021 included and downloaded on 2 Jun 2021 via gisaid.org (GISAID: EPI_SET_230724tv). To view the contributors of each sequence associated with the metadata we used, visit https://doi.org/10.55876/gis8.230724tv. Proprietary airline data are commercially available from OAG and IATA databases (https://www.iata.org/). All other data used in the study are publicly available online. Channel Tunnel data were obtained from https://www.eurotunnelfreight.com/fr/2021/01/trafic-navettes-du-mois-de-decembre-2020/, ferries data were obtained from https://www.gov.uk/government/statistical-data-sets/sea-passenger-statistics-spas, COVID-19 cases were obtained from https://github.com/CSSEGISandData/COVID-19, COVID-19 deaths were obtained from https://www.ecdc.europa.eu/en/publications-data/data-national-14-day-notification-rate-covid-19, Alpha cases in the UK were obtained from https://assets.publishing.service.gov.uk/media/6059e4ad8fa8f545d5c339fc/Variants_of_Concern_VOC_Technical_Briefing_7_England.pdf.

## Code availability

Source codes to reproduce the results of this study are publicly shared on zenodo[80] and on github (https://github.com/EPIcx-lab/COVID-19/tree/master/Adherence_and_sustainability).

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

## Acknowledgements

The authors gratefully acknowledge all data contributors, i.e., the Authors and their Originating laboratories responsible for obtaining the specimens, and their Submitting laboratories for generating the genetic sequence and metadata and sharing via the GISAID Initiative, on which this research is based. We acknowledge financial support by the Municipality of Paris through the programme Emergence(s) to C.P. and B.F.; Cariparo Foundation through the program Starting Package to C.P.; Department of Molecular Medicine through the program SID from BIRD funding to C.P.; the Agence Nationale de la Recherche project DATAR-EDUX with grant agreement ANR-19-CE46-0008-03 to V.C.; ANRS–Maladies Infectieuses Émergentes project EMERGEN (ANRS0151) to V.C.; EU Horizon 2020 grants MOOD with grant agreement H2020-874850 (publication cataloged as MOOD096) to V.C., C.P., P.Y.B., M.U.G.K., P.L. and RECOVER (H2020- 101003589) to V.C.; the ERC grant EvoComBac (949208) to F.B.; ERC grant ReservoirDOCS (725422) to P.L.; Marie Sklodowska-Curie action (MSCA) grant PolyPath (844369) to P.C.; Institut des Sciences du Calcul et de la Donnée (ISCD).

## Author contributions

V.C., F.B., C.P., and P.-Y.B. conceived and designed the study. B.F., P.-Y.B., and C.P. developed the international dissemination model. P.C., and F.B., developed the autochthonous model A. C.E.S. and V.C. developed the autochthonous model B. M.U.G.K. and P.L. critically commented on the model ingredients and assumptions. V.C., F.B., C.P., and P.-Y.B. wrote the original draft. All authors discussed the results, edited the manuscript, and approved its final version.

## Competing interests

The authors declare no competing interests
