## [Peer Review File · Nature Communications]

Drivers and impact of the early silent invasion of SARS-CoV-2 AlphaREVIEWER COMMENTS

Reviewer #1 (Remarks to the Author):

The manuscript provides an in-depth analysis of the silent spread of the SARS-CoV-2 Alpha variant, integrating international mobility, local epidemic growth, and genomic surveillance into a comprehensive Bayesian framework. The paper is well-structured and addresses a critical issue in pandemic management – understanding the factors influencing the silent spread of a virus and its subsequent impact on epidemic progression. The use of many different dataset is a highlight, but the sophisticated modeling techniques worry me a bit on the model reliability.

The paper overall reads well. The abstract provides a concise and clear overview of the study, its methodologies, and key findings. It highlights the importance of understanding silent spread and its implications for future pandemic responses. The introduction effectively sets the stage for the study, providing context on the Alpha variant and the challenges associated with its silent spread. The result section clearly states the factors contributing to the spread of alpha, and the local dynamics that affected the impact of silent spread.

Nevertheless, I still have the following fundamental questions:

1. It could benefit from a more explicit statement of the research objectives and how this study aims to address the existing gaps in knowledge, especially given the vast existing literature on COVID-19.
2. Related to my first comment, the author should illustrate why the paper is still timely and of relevance, standing as of now in 2023. And how this retrospective study for 2020 spread is still important despite the COVID-19 research fatigue in the academic community.
3. For the detailed method, the model includes various fixed parameters and assumptions. While the authors have conducted sensitivity analysis on the fixed parameters to test the robustness, the justification on model structure and assumptions (e.g. additive component, exponential rating, etc) are less elaborated. I appreciate that the posterior predictive check is conducted, but is that enough for such a complicated model? It is very hard to validate this model, and to certain extent it requires a leap of faith. Also, the method consists of many sub-modules that are cited from other studies. Are all the assumptions from different references even compatible, or appropriate? e.g. Autochthonous model B is taken from a previous work for France, and International dissemination model is primary for UK.

Reviewer #2 (Remarks to the Author):

The manuscript "Drivers and impact of the early silent invasion of SARS-CoV-2 Alpha" reports findings from mathematical modelling of Alpha dissemination outside of the UK in the late 2020. The models take into account travel flows from the UK and destination countries, sequencing coverage, and estimated transmission dynamics in the UK and outside. The paper is generally well written; while a number of simplifying assumptions is used (such as that the introductions can only happen from the UK even though Alpha was seeded in other countries well before the end of the observation period of December 31st 2020), the modeling assumptions are mostly well-described and tested in sensitivity analyses. I have the following comments/concerns:

1. Given that Alpha did not demonstrate immune-evasive properties, such as some of the other variants emerging at the time, it is reasonable to assume that the success of Alpha in "destination" countries would have been affected by seroprevalence level in those countries in the aftermath of the first wave. Given that there has been a great variation in the scale of the first SARS-CoV-2 wave in the considered countries, this factor needs to at least be discussed as it is currently not mentioned.
2. It wasn't clear to me why the sequencing coverage was considered as the number of collected sequences divided by the number of reported cases regardless of the date of submission (line 89). This introduces bias with regards to the general sequencing capacity (some countries might have resources to do more retrospective sequencing than others) but also in terms of dissemination

patterns, since only the actual sequencing coverage at the time of the introduction matters.

3. Some of the findings were not put in context – line 124 reports the seeding date in the UK but doesn't compare it to other published data (i.e. Hill et al 2022).

4. The authors notice that the incidence estimates for the US didn't agree with the data and they rightly suggest that it might be due to heterogeneity between states. Have attempts been made to fit the model to a single state (i.e. NY for which comprehensive data available)?

5. Some of the conclusions are overstated: "Yet our analysis showed that traveling flows remained the main drivers of viral spread, in agreement with other works" (line 239) suggests that multiple drivers were "compared", however, even though the model "relied" on the travel data, no comparisons were made. Another example is that the authors state "limiting importations through border control could contribute to mitigation"; however, no formal testing of how the further reduction in travel would have affected seeding in the destination countries was done. Many other studies showed that even when travel bans were in place before a variant emergence, they were still successfully disseminated (i.e. Gamma in the US thought travel ban from Brazil in early 2020).
Minor additional comments:

1. Fig 3B/C – the order of countries is unclear, alphabetic would be better for easier location of the country of a graph.

2. It's not always clear which sub-set of countries (73, 24, or 6) was used for which analyses – i.e. line 151.

3. It was not immediately clear (before reading the methods) whether R_t estimates are for SARS-CoV-2 in general or Alpha only.

Reviewer #3 (Remarks to the Author):

In this study, authors evaluated the early dissemination SARS-CoV-2 Alpha Variant of Concern (VoC) in the UK using a Bayesian framework, to reconstruct its spatial and temporal dynamics across countries using different sources of data such as international mobility data, and epidemiological and genomic surveillance data. Authors argue that retrospectively understanding the dynamics of VoC, and in particular the Alpha case-study, can facilitate better public health responses in the future.

The article reads well and its motivation, objectives, methods and conclusions are clearly explained and justified. The modeling approach is generally well explained and, although it requires many assumptions -some of them tested for sensitivity and/or reasonably tenable- it is well justified and validated with empirical observations and consistent with available evidence.

The overall conclusion of the analysis is that Alpha cases were imported but remained undetected for long periods of time across many countries, this delays in detection correlating mostly with sequencing coverage. Mobility of individuals is uncovered as the major driver of importation - consistent with the theory of communicable diseases epidemiology- , a process that can be misrepresented using genomic surveillance without accounting for potential bias.

While this work reinforces the idea that adequately designed genomic surveillance is critical in early detection of emerging pathogens, the idea of more granular surveillance equals better response is not novel and not always necessarily practical in public health, particularly when prospective surveillance entails many challenges as the impact of emerging clones/pathogens remains unknown. Nevertheless, the work adds value as a potentially integrative source of surveillance evidence and methods, embedding genomic surveillance and accounting for potential bias for its interpretation.

Methods:

Line 309. What's the rationale for inferring a single empirical random delay, instead of a modeled one, for example? What are the consequences of this imputation in terms of the robustness of the model, given the small percentage of missingness (3%)?

I understand that submission date (submission to GISAID database), can be interpreted as when

the information was available, versus collection date, which is when would be the earliest information available in a "better performing" surveillance system given importation. I would suggest to describe clearly this interpretation in the methods for better understanding of the approach by public health practitioners (or the one that authors make for their conclusion if I am incorrect)

Line 322. Was there any adjustment for the ascertainment/reporting bias on the COVID-cases/deaths? How would that change the subsequent estimates in, for example, the reproduction number or force of infection?

Line 339. Why the 95% reduction assumption in rail passenger numbers? Probably not very relevant, but does not seem clearly justified.

Line 349 and Table S1. Have the authors evaluated how would include uncertainty in the estimates (i.e., on the 7 days delay between infection and reporting, or on the survey estimates of alpha rates)? Google data aggregator represents cases reported thus, might introduce some bias regarding cases ascertainment and reporting. If so, how would this affect the model estimates? Also I might be missing something, but why use this source of data for virological surveys and not the incidence of cases obtained for sequencing coverage?

Line 386. What is the rationale for assuming that the first GISAID submitted Alpha case is imported? Is there a way to test this (i.e. meta data) and how this assumption might change the conclusions?

Results

Fig 1 A. This plot is difficult to read, as it is difficult to follow many lines of the same color that cross over each other; thus difficult to see the trend. Is there a better way to show the trends?

Fig 1 B. Given that early detection correlates with higher (log) coverage and higher (log) flow, shouldn't the trend highlighted be $y=x$? The dispersion seems to be wider on the sequencing dimension rather than on the travel flow.

Fig 1 D. It is unclear what is represented here. Is this plot representing the distributions of the median delay distribution by country or the joint distribution of delays for all countries?

Line 103. Which period? September -31st December 2020?

Line 108 What is the definition of the early dissemination period? How might this assumption change the model results? (i.e., the first detected cases was not imported, or was imported from other country than UK)

Fig 2 C Is the gray coloured ribbon the uncertainty range? How was this computed?

Line 155. I understand that 70 days is the higher bound? Is there a way to summarize the distribution more clearly (i.e, a central estimate and range) if this is informative?

Fig 3D y axis label (silent spread) is defined as mean/median number of days between estimated introduction and detection? I think the unit (days) should be included in the plot and clarify the definition.

Line 184. Is the rate estimated? Is it similar across countries (except the US)?

Supplementary

Fig S1. Date unit is weeks?

Discussion

Line 241. I believe this is a major claim in terms of usefulness of the approach, but the statement feels a bit vague and nonspecific. The discussion might benefit from more specific justification, particularly when contextualizing with existing work describing how to design and improve this

type of surveillance on practical terms.

Line 249. This statement, while true, resonates with some of the problems stated in the beginning of the review. Retrospectively evaluating the need of detection of a specific strain that later on becomes relevant is straightforward. However, if other sources of evidence beyond case surveillance are not available on time, public health bodies will still struggle to make complex decisions. That feels that has to be somehow addressed, otherwise becomes a simplified interpretation and not so useful if the work intends to improve response in the future.

Line 301. I believe this conclusion from the work is very useful and merits more discussion, as provides evidence that even when an emerging pathogen is not reported, the fact that is not circulating is not necessarily the explanation, but likely it is due to surveillance systems not being able to detect it, specially when major known drivers are indeed in place (i.e., travel flow etc).

Point-by-point reply of Drivers and impact of the early silent invasion of SARS-CoV-2 Alpha

We are grateful to the Reviewers for their feedback and careful reports. We are glad that they recognised that the study addresses a critical issue, it is generally well written, and motivation, objectives, methods and conclusions are clearly explained and justified. To address their concerns we have revised the study. Detailed explanation of the changes made in the manuscript in response to the Reviewers' comments are reported in the point-by-point reply below.

Answers to the comments of Reviewer 1

The manuscript provides an in-depth analysis of the silent spread of the SARS-CoV-2 Alpha variant, integrating international mobility, local epidemic growth, and genomic surveillance into a comprehensive Bayesian framework. The paper is well-structured and addresses a critical issue in pandemic management – understanding the factors influencing the silent spread of a virus and its subsequent impact on epidemic progression. The use of many different dataset is a highlight, but the sophisticated modeling techniques worry me a bit on the model reliability.

The paper overall reads well. The abstract provides a concise and clear overview of the study, its methodologies, and key findings. It highlights the importance of understanding silent spread and its implications for future pandemic responses. The introduction effectively sets the stage for the study, providing context on the Alpha variant and the challenges associated with its silent spread. The result section clearly states the factors contributing to the spread of alpha, and the local dynamics that affected the impact of silent spread.

Nevertheless, I still have the following fundamental questions:

1. It could benefit from a more explicit statement of the research objectives and how this study aims to address the existing gaps in knowledge, especially given the vast existing literature on COVID-19.

We have revised the Introduction to more explicitly state both the research gaps the study aims at addressing and its research objectives.

More precisely the paragraph describing the knowledge gaps regarding silent spread and its public health significance has been modified and expanded. It now reads as follows:

-- Phylodynamics analysis and modeling studies revealed that silent spread occurred for early SARS-CoV-2 lineages and subsequent variants of concern (VOCs) (12–20). This has sparked a public health debate. The efforts to contain a variant at the source are ineffective if they come too late, when the virus is already spreading cryptically out of the source.

Interventions aiming at mitigation or delay may instead have an impact depending on the extent and duration of silent dissemination at the time they are implemented (21,22). Recent work addressed the minimal sequencing coverage to detect a variant early enough for an effective response, and proposed modeling tools for risk assessment (23–27). However, the complex interplay of factors determining the duration of silent propagation remains poorly understood.

We have then added a paragraph at the end of the introduction to explicitly state the objective of the work

-- Here we used a Bayesian model to retrospectively reconstruct the initial international dissemination of Alpha from 20 Sep 2020 to 31 Dec 2020 out of the UK. By leveraging diverse sources of data for the temporal and geographical change in international travel, sequencing coverage and local epidemic growth, we show that these factors, together with the effect of the international VOC alert on surveillance, drove the duration of Alpha silent spread.

2. Related to my first comment, the author should illustrate why the paper is still timely and of relevance, standing as of now in 2023. And how this retrospective study for 2020 spread is still important despite the COVID-19 research fatigue in the academic community.

SARS-CoV-2 VOCs still represent a serious threat as demonstrated by the attention dedicated by surveillance authorities to the monitoring of new variant emergence. The need for the design of sustainable, still effective, surveillance protocols call for a better understanding of the impact of surveillance efforts on our capacity to detect and respond to new variants in time. More broadly, retrospective analyses of the spread of SARS-COV-2 VOCs provide both new methodologies and lessons learnt for the response to future pandemic threat.

We have expanded the Discussion including these considerations. At the beginning of the Discussion we have added the sentence:

-- Retrospective analyses of the past dissemination of VOCs can provide epidemiological knowledge that enable us to better respond to future viral emergence events.

A paragraph in the Discussion now reads:

-- ... The Alpha experience shows the importance of designing sequencing protocols able to balance sustainability and detection capacity by meeting the minimal requirements of sequencing extent and reporting delay - e.g. sequencing 0.5% of cases with a turnaround time smaller than 21 days as previously proposed 23 -, and by leveraging information from multiple sources, including wastewater and animal surveillance (56,57). Importantly, this study also highlights that the knowledge of surveillance extent and protocol adopted by countries is key to real-time data analysis to better assist risk assessment and intervention planning. This would be facilitated by the widespread adoption of pre-established surveillance protocols.

We have also added the following concluding sentence:

-- ... Taken together these findings provide lessons learnt for the future management of SARS-CoV-2 variants. Beyond that, retrospective reconstructions of SARS-CoV-2 spread are essential to improve computational modeling and public health knowledge to better guide the response to future spread of viruses with zoonotic and pandemic potential.

3. For the detailed method, the model includes various fixed parameters and assumptions. While the authors have conducted sensitivity analysis on the fixed parameters to test the robustness, the justification on model structure and assumptions (e.g. additive component, exponential rating, etc) are less elaborated. I appreciate that the posterior predictive check is conducted, but is that enough for such a complicated model? It is very hard to validate this model, and to certain extent it requires a leap of faith.

The international dissemination model (IDM) has a very simple structure depending on 3 unknown parameters in all : 2 rates of growth for the alpha variant in the source country (UK) and a factor for the increase in detection after the December 18th alert. It is based on minimal assumptions - i.e. proportionality rules - to combine these parameters with sequencing (GISAID), incidence (Johns Hopkins CSSE) and international travel (air traffic from IATA combined with other official sources) to compute expected introduction rates. We view it as a strength of our work that this direct approach fits the data at all. We agree that this cannot be a validation, but believe that it certainly lends some credibility to the model.

Regarding the hypotheses:

- exponential growth is a very standard description of epidemic growth;
- proportionality of importations to air traffic is the standard assumption to study international spread of diseases;
- the “additive” notation on the rate ($\exp \sum_{u \leq t} r(u)$) translates to a “two-slopes” exponential incidence curve ($i(t) = \exp(r_0 t)$ before the lockdown at date T and $i(t) = \exp(r_0 t + r_1(t - T))$ afterwards) since the rates $r(u)$ take value r_0 before the UK lockdown and value r_1 afterwards;
- the “time to event” models are standard statistical models for rate based descriptions.

As noted by the Referee, we have provided several predictive checks that show good agreement between the IDM predictions and the GISAID metadata, and with the autochthonous spread models and virological data.

We have modified the Methods section to better clarify the overall structure of the model, to better explain the additive notation of the incidence rate as a way to capture the two-slope behavior, and to better justify the parameters and assumptions chosen for the sensitivity analysis, i.e. the ones mostly affected by model uncertainty.

Changes were done throughout the Methods section (lines 430-433, 438-442, 445-448, 453-454, and 464-466, revised manuscript). In particular, in lines 445-448 we added the following sentence to better explain the additive notation:

-- The daily exponential growth rate in the UK was considered piecewise constant, r_0 up to Nov 5th, 2020, when the UK entered a lockdown, and r_1 afterwards. In other words, $inc_{UK}(t)$ was a “two-slope” exponential, growing as $exp(r_0 t)$ before Nov 5 and as $exp((r_0 + r_1)t)$ afterwards.

Also, the method consists of many sub-modules that are cited from other studies. Are all the assumptions from different references even compatible, or appropriate? e.g. Autochthonous model B is taken from a previous work for France, and International dissemination model is primary for UK.

The international dissemination model (IDM) and the autochthonous transmission models describe subsequent processes: first the exportation of cases out of the UK; then the local transmission following importation in the destination countries. In our study, these steps are linked by using the output of the first as input to the second, and this serves to reinforce the plausibility of the IDM analysis and to explain observed differences in local situations.

It is indeed interesting to question whether the approaches are “compatible” with each other. Obviously, all models allow for a time-dependent exponential spread of COVID-19, the Autochthonous model A and B using mechanistic descriptions of transmission chains and the IDM a phenomenological description. In this way, their behavior is “compatible”. Also, model parameters, calibrated using available evidence, are similar when they can be formulated alike in different models: for example, the incubation period is 5 days in the IDM and 5.2 days in the autochthonous model B; the generation time in autochthonous transmission model A is 6.5 days and 6.6 days in model B.

We realized that the reference for the choice of the incubation period was missing and we have now added it (line 454, revised manuscript). In addition, for completeness, we have reported in lines 469-471 (revised manuscript) the values of some key parameters of the autochthonous model B - the parameters listed in full in the references cited. It is important to note that model A and B have a different model structure. The comparison between two different models confirm the robustness of the findings.

Answers to the comments of Reviewer 2

The manuscript “Drivers and impact of the early silent invasion of SARS-CoV-2 Alpha” reports findings from mathematical modelling of Alpha dissemination outside of the UK in the late 2020. The models take into account travel flows from the UK and destination countries, sequencing coverage, and estimated transmission dynamics in the UK and outside. The paper is generally well written; while a number of simplifying assumptions is used (such as that the introductions can only happen from the UK even though Alpha was seeded in other countries well before the end of the observation period of December 31st 2020), the modeling assumptions are mostly well-described and tested in sensitivity analyses. I have the following comments/concerns:

1. Given that Alpha did not demonstrate immune-evasive properties, such as some of the other variants emerging at the time, it is reasonable to assume that the success of Alpha in “destination” countries would have been affected by seroprevalence level in those countries

in the aftermath of the first wave. Given that there has been a great variation in the scale of the first SARS-CoV-2 wave in the considered countries, this factor needs to at least be discussed as it is currently not mentioned.

In the first part of the paper (International dissemination model), we focus on the date of importation of the Alpha variant and its detection depending on sequencing effort. This is done regardless of whether it leads to secondary cases or not, so that the immunity level of the target population would not change the findings.

We acknowledge that the immune status of the population is relevant in the second part of the paper, where we modeled autochthonous spread. In the Autochthonous model B immunity is explicitly accounted for. In the autochthonous model A the transmission potential of Alpha is proportional to the country-specific time-varying reproductive ratios computed from empirical data. This accounts for all factors affecting transmission, e.g. the level of mixing, as related to local interventions, and population immunity. In each country, we assumed that the relative advantage of Alpha was the same, i.e. not dependent on the population immunity (in line with the fact that Alpha did not present much immune-evasive properties). We have now made this assumption explicit in the text in line 520 (revised manuscript).

2. It wasn't clear to me why the sequencing coverage was considered as the number of collected sequences divided by the number of reported cases regardless of the date of submission (line 89). This introduces bias with regards to the general sequencing capacity (some countries might have resources to do more retrospective sequencing than others) but also in terms of dissemination patterns, since only the actual sequencing coverage at the time of the introduction matters.

In Fig 1. B and C we have, indeed, computed the sequencing coverage as described by the Reviewer. In the International Dissemination Model (IDM) we accounted for both the time of collection and submission of a sequence to simultaneously fit the date of collection and submission of the first collected sequence. Indeed, the quantity $\pi(t - u; u) = \frac{n_t}{N_u}$ corresponds to the fraction of all samples collected at date u that are submitted at date t , using the actual coverage at each date.

In general, it is true that the collection-submission delay is very broad and there is no criterion to set a cut off on it to distinguish between real-time and retrospective surveillance when evaluating sequencing coverage. It is also not possible to estimate the magnitude of this effect to understand whether it is relevant. We have listed this point among the limitations in the Discussion by adding the sentence

-- Third, it is not possible to set a cut off between real-time and retrospective surveillance when computing sequencing coverage from GISAID metadata. The computation of sequencing coverage being affected by retrospective surveillance could potentially overestimate the extent of the real time genomic surveillance.

3. Some of the findings were not put in context – line 124 reports the seeding date in the UK but doesn't compare it to other published data (i.e. Hill at all 2022).

We thank the Reviewer for pointing this out. We added the missing reference in line 133 (revised manuscript).

4. The authors notice that the incidence estimates for the US didn't agree with the data and they rightly suggest that it might be due to heterogeneity between states. Have attempts been made to fit the model to a single state (i.e. NY for which comprehensive data available)?

We thank the Reviewer for this suggestion. We attempted to compare the results from the autochthonous model on two individual states, California and Florida, for which data were available in the reference that we used (Washington et al. Cell 2021), and for New York City, for which data were publicly available (<https://github.com/nychealth/coronavirus-data/tree/master/variants>). These locations were the port of entry of Alpha into the US, with early reported Alpha cases linked directly to the UK (Washington et al. Cell 2021, Alpert et al Cell 2021).

As for the USA as a whole, the autochthonous model A was fed with importation fluxes estimated from the international dissemination model and we compared the model-predicted number of Alpha cases with the empirical estimates. For California and New York City, the comparison between model and empirical estimates follows a trend similar to European countries. Florida registered a high proportion of Alpha cases. Such a high level of Alpha circulation can be compatible with model predictions in a scenario of early Alpha introduction, i.e. introduction dates close to the lower bound of the range predicted by the model.

We present these results in Supplementary Fig. 4 and in section "Local dynamics in the USA at a finer spatial scale" of the Supporting Information. We also discuss these results in the main paper (lines 199-202).

5. Some of the conclusions are overstated: "Yet our analysis showed that traveling flows remained the main drivers of viral spread, in agreement with other works" (line 239) suggests that multiple drivers were "compared", however, even though the model "relied" on the travel data, no comparisons were made. Another example is that the authors state "limiting importations through border control could contribute to mitigation"; however, no formal testing of how the further reduction in travel would have affected seeding in the destination countries was done. Many other studies showed that even when travel bans were in place before a variant emergence, they were still successfully disseminated (i.e. Gamma in the US thought travel ban from Brazil in early 2020).

We have revised the text to make it more balanced.

In reference to the first point, the international dissemination model fits the data well. This suggests that the model ingredients - specifically, the input data related to international travel and sequencing coverage - effectively capture the key factors driving both propagation and detection. In addition, we show in Fig. 4 B that the number of cases in the six countries considered in the in-depth analysis increases with the flux of travelers from the UK. In

response to the Reviewer's feedback, we have modified the sentence in the Discussion to soften the claim while simultaneously being more specific. It now reads:

-- Yet the good fit obtained with the international dissemination model and the increase of Alpha epidemic size with traveling flow (Fig. 4B) both suggest that traveling flows are a driver of viral spread, in agreement with other works

Concerning the second point raised by the Reviewer. We have not formally tested how a change in importations would affect the VOC frequency in a given country. We have thus made it clear that our considerations are more an hypothesis. We have also put the sentence into context of previous work including the work cited by the Reviewer

-- We can thus hypothesize that limiting importation early could act synergistically with local restrictions to limit the size of the VOC epidemic. Still, we expect that the fine tuning between different factors (e.g. quality and extent of border controls and timing of their implementation (22,52,53)) can affect the impact of travel restrictions.

Minor additional comments:

1. Fig 3B/C – the order of countries is unclear, alphabetic would be better for easier location of the country of a graph.

We thank the Reviewer for the suggestion. We have replaced Fig. 3B/C with a new version with countries in alphabetical order.

2. It's not always clear which sub-set of countries (73, 24, or 6) was used for which analyses – i.e. line 151.

To sum up, the international dissemination model considers the 73 countries which submitted at least one sequence on GISAID during the study period (between September and December 2020). The date of submission (and respective date of collection) for the 24 countries for which at least an Alpha sequence was reported were fitted by the model, the absence of reported sequences in the other countries was also accounted for with statistical censoring.

Local transmission in the six countries for which early virological investigations were available were analyzed in detail in the last part of the Results section.

We are aware that the use of different sets of countries may be confusing. We have revised the text to clearly specify the number of countries considered - Line 110 and line 163 of the revised manuscript.

3. It was not immediately clear (before reading the methods) whether R_t estimates are for SARS-CoV-2 in general or Alpha only.

To avoid any confusion this is now clearly indicated in the axis labels of the plot of Fig. 4 C and D

Answers to the comments of Reviewer 3

In this study, authors evaluated the early dissemination SARS-CoV-2 Alpha Variant of Concern (VoC) in the UK using a Bayesian framework, to reconstruct its spatial and temporal dynamics across countries using different sources of data such as international mobility data, and epidemiological and genomic surveillance data. Authors argue that retrospectively understanding the dynamics of VoC, and in particular the Alpha case-study, can facilitate better public health responses in the future.

The article reads well and its motivation, objectives, methods and conclusions are clearly explained and justified. The modeling approach is generally well explained and, although it requires many assumptions -some of them tested for sensitivity and/or reasonably tenable- it is well justified and validated with empirical observations and consistent with available evidence.

The overall conclusion of the analysis is that Alpha cases were imported but remained undetected for long periods of time across many countries, this delays in detection correlating mostly with sequencing coverage. Mobility of individuals is uncovered as the major driver of importation - consistent with the theory of communicable diseases epidemiology- , a process that can be misrepresented using genomic surveillance without accounting for potential bias.

While this work reinforces the idea that adequately designed genomic surveillance is critical in early detection of emerging pathogens, the idea of more granular surveillance equals better response is not novel and not always necessarily practical in public health, particularly when prospective surveillance entails many challenges as the impact of emerging clones/pathogens remains unknown. Nevertheless, the work adds value as a potentially integrative source of surveillance evidence and methods, embedding genomic surveillance and accounting for potential bias for its interpretation.

Methods:

Line 309. What's the rationale for inferring a single empirical random delay, instead of a modeled one, for example? What are the consequences of this imputation in terms of the robustness of the model, given the small percentage of missingness (3%)?

Given the small percentage of missing records, imputation may indeed have little impact on the overall results. We preferred inferring an empirical delay distribution over a modeled one because it may be highly challenging to capture spatiotemporal variation by a statistical model. The use of the empirical distribution requires minimal assumptions. We have better justified this choice, by adding on the method section the sentence

-- We resorted to imputation instead of inferring a statistical model because of the small percentage of missing records. In addition, the strong spatiotemporal variations displayed by the data could be hard to capture by a statistical model.

I understand that submission date (submission to GISAID database), can be interpreted as when the information was available, versus collection date, which is when would be the earliest information available in a “better performing” surveillance system given importation. I would suggest to describe clearly this interpretation in the methods for better understanding of the approach by public health practitioners (or the one that authors make for their conclusion if I am incorrect)

The Reviewer is right that taking the collection date as the detection date is an optimistic assumption. In that, the underlying hypothesis is that surveillance authorities are notified right after the sequence is collected. This could be the case in a situation of alert and heightened surveillance. Conversely, assuming surveillance authorities are notified of an Alpha case once the sequence is submitted to GISAID would imply a longer delay from importation to detection, i.e. longer silent spread. The median time from collection to submission for Alpha sequence during the period was indeed 27 days, as discussed in the Results. We have already discussed this issue in the limitation paragraph in the Discussion section. We have now made this point clearer by adding a sentence to the Methods section, as suggested by the Reviewer:

-- Assuming that detection occurs at the time of sample collection corresponds to the optimistic hypothesis that surveillance authorities are informed right after a sample is collected.

Line 322. Was there any adjustment for the ascertainment/reporting bias on the COVID-cases/deaths? How would that change the subsequent estimates in, for example, the reproduction number or force of infection?

The analysis of the local spread following importation in the six focal countries (last subsection of the Result section, Fig. 4) is agnostic to assumptions regarding the reporting ratio. The reproductive ratio - which rules the force of infection - is computed from the ratio between deaths in week w and week $w + 1$. This ratio remains the same regardless of the reporting rate, provided this rate is constant in time.

The comparison of local cases predicted by the autochthonous models vs. the ones estimated by surveillance required no a-priori assumptions on the reporting ratio. Our results show that modeled and reported cases follow the same trend, with modeled cases being roughly twice the reported cases, which could be attributed to the reporting ratio.

The comparison between the modeled and reported Alpha epidemic in the UK, shown in Fig. 2 C, is based on incidence trends. The number of cases has been rescaled to the sum over the study period in order to compare the growth regardless of the detection rate.

We have noticed that this was not always explicitly stated. We have revised the text where needed, i.e. :

-- Note that the calculation of R_t in this way is robust to under reporting biases, provided that the reporting ratio does not change substantially over the period.

In contrast to the analyses of local transmission, the international dissemination model relies on the assumption of case detection rate (50% of imported cases detected as COVID-19 cases). An alternative value was tested in the sensitivity analysis.

Line 339. Why the 95% reduction assumption in rail passenger numbers? Probably not very relevant, but does not seem clearly justified.

This is based on the following source

<https://www.breakingtravelnews.com/news/article/eurostar-left-battling-for-survival-following-covid-19-slump/>

We have added the citation in the text in line 391 (revised manuscript).

Line 349 and Table S1. Have the authors evaluated how would include uncertainty in the estimates (i.e., on the 7 days delay between infection and reporting, or on the survey estimates of alpha rates)? Google data aggregator represents cases reported thus, might introduce some bias regarding cases ascertainment and reporting. If so, how would this affect the model estimates?

Model estimates are agnostic to biases in the ratio of reporting of Alpha cases in the six destination countries. Model estimates are based on the flow of imported cases predicted by the international dissemination model and on the estimated reproductive ratio, which is unaffected by the reporting ratio, under the assumption this is constant in time - see reply to the previous point. The comparison between model and reported cases estimates in Fig. 4A is also insensitive to a-priori assumptions on the reporting ratio - see reply to the previous point. It is true, however, that the plot of Fig. 4A relies on the assumption of a 7-day delay in reporting. Delay in reporting may vary from one country to another. Also some countries report cases by date of testing, others by date of notification, and smoothing is applied in some surveillance reports. To account for this layer of uncertainty we have added to the error bar in Fig. 4A a third layer corresponding to the assumed delay of reporting. Instead of only assuming a 7 day delay, we extended the error to show the predicted number of Alpha infections if the delay is 4 days (leading to greater modeled number of Alpha infections) or 10 days (leading to smaller modeled number of Alpha infections). The text was updated to reflect this change in lines 539-541 (revised manuscript).

Also I might be missing something, but why use this source of data for virological surveys and not the incidence of cases obtained for sequencing coverage?

We originally used the Google data aggregator which gathers data from various sources, often specific governmental sources for each country. For consistency, we now use the daily cases from the Center for Systems Science and Engineering at Johns Hopkins University (CSSE) as used to compute the sequencing coverage. We have updated Figure 4 and Supplementary Table 1. Although the numbers differ slightly, this does not have any impact on our results.

Line 386. What is the rationale for assuming that the first GISAID submitted Alpha case is imported? Is there a way to test this (i.e. meta data) and how this assumption might change the conclusions?

We relied on a risk assessment report on reported Alpha cases in European countries by end of December 2020, published by ECDC (<https://www.ecdc.europa.eu/en/publications-data/covid-19-risk-assessment-spread-new-sars-cov-2-variants-eueea>), and on the study by O'Toole et al which tracked the first reported Alpha cases worldwide (<https://www.ncbi.nlm.nih.gov/pmc/articles/PMC8176267/pdf/wellcomeopenres-6-19025.pdf>), and gathered information regarding the surveillance protocols in place in the different countries. These two sources show that for the majority of countries first reported cases were traveling cases, consistent with the fact that sequencing of traveling cases was prioritized. These sources are already cited in the Methods section. We have now explicitly discussed this hypothesis and cited the two sources already in the Results section to make this more explicit early on:

-- Thus, we assumed that before the end of December, the first detected cases were traveling cases (4,32) and dissemination was at its early stage, i.e. traveling cases were traveling out of the UK.

Note that relaxing this assumption in our international dissemination model means that introduction from the UK would have occurred even before. Consequently, longer duration of silent spread would be expected, leading to estimating a quicker growth of cases in the UK. We note that this would not be compatible with the Alpha incidence data in the UK and literature estimates of Alpha growth rates.

Results

Fig 1 A. This plot is difficult to read, as it is difficult to follow many lines of the same color that cross over each other; thus difficult to see the trend. Is there a better way to show the trends?

We have colored the lines using four colors according to the month in which air travel was at its maximum value. We believe that the readability of the plot is now improved.

Fig 1 B. Given that early detection correlates with higher (log) coverage and higher (log) flow, shouldn't the trend highlighted be $y=x$? The dispersion seems to be wider on the sequencing dimension rather than on the travel flow.

We chose to keep the line $y = -x$ because it corresponds to a theoretical result under the assumption of exponential growth in the UK and traveling flow constant in time: the date of collection depends on $\log(\text{international traffic}) + \log(\text{sequencing coverage})$ and therefore two countries on a line with slope -1 should have similar dates of collection.

We have more clearly specified that in the caption of the figure and added a paragraph in the Supplementary Information to better explain the theory ("Arrival times, number of passengers and sequencing coverage").

Fig 1 D. It is unclear what is represented here. Is this plot representing the distributions of the median delay distribution by country or the joint distribution of delays for all countries?

We represent here the distribution of delays over all countries, for Alpha and non-Alpha cases. We have clarified this in the caption of Fig. 1.

Line 103. Which period? September -31st December 2020?

We considered all dates of collection of the first submitted sequence in countries outside the UK before 31 Dec 2020 and we included in the fit also the date of the earliest reported Alpha sequence in the UK (20 Sep). Therefore, fitted dates range between 20 Sep to 31 Dec. This is already detailed in the methods. To make this clear in the Results section we have modified the sentence as follows:

-- We developed the Alpha international dissemination model to fit the date of first detection and the corresponding date of submission between the beginning of September and end of December in the 73 countries contributing to GISAID during the period.

Line 108 What is the definition of the early dissemination period? How might this assumption change the model results? (i.e., the first detected cases was not imported, or was imported from other country than UK)

We made the hypothesis that before the end of December 2020 Alpha was at its earlier dissemination stage, i.e. as correctly noted by the Reviewers, first detected cases were imported and importations originated directly from the UK. As mentioned in the point above, we have change the sentence as follow:

-- Thus, we assumed that before the end of December, the first detected cases were traveling cases (4,32) and dissemination was at its early stage, i.e. traveling cases were traveling out of the UK.

Fig 2 C Is the gray coloured ribbon the uncertainty range? How was this computed?

The gray coloured ribbon is the 95% credible interval rescaled to the sum of the cases in the median scenario. We now clarify it in the figure caption.

Line 155. I understand that 70 days is the higher bound? Is there a way to summarize the distribution more clearly (i.e, a central estimate and range) if this is informative?

The silent spread, defined as the time between the median estimated date of introduction and the median estimated date of first detection, ranged from 12 days to 70 days across countries, with a median at 40 days.

We have added the distribution of the silent spread across countries in the Supplementary Fig. 3)

Note that Fig. 3 D has been revised to include only those countries for which both the introduction and the detection were predicted to occur before early January to avoid model extrapolations beyond the period of validity of the model.

Fig 3D y axis label (silent spread) is defined as mean/median number of days between estimated introduction and detection? I think the unit (days) should be included in the plot and clarify the definition.

We thank the Reviewer for pointing this out. We have added the unit on the y-axis.

Line 184. Is the rate estimated? Is it similar across countries (except the US)?

In Fig. 4 A we have made no a-priori assumption on the reporting rate. The figure shows on the log scale three gray lines corresponding to $y=x+A$ where A indicates 25%, 50% and 100% reporting rate, respectively. With the exception of the US, which is out of trend, dots roughly lie along the gray line that corresponds to a 50% reporting rate, therefore we concluded that the model reproduces the trend observed in real data with a case ascertainment fraction of roughly 50%. A precise estimation of the reporting rate is beyond the scope of this analysis.

We acknowledge that the sentence in the original version of the paper may give a too precise impression. Therefore we have rewritten it as follows:

-- The model reproduced the same trend of the observed Alpha cases with a case ascertainment fraction around 50% (Fig. 4A), ...

Supplementary

Fig S1. Date unit is weeks?

Yes, in Supplementary Fig. 1, each row and each column represent a week. We have now made it clear on the axis labels

Discussion

Line 241. I believe this is a major claim in terms of usefulness of the approach, but the statement feels a bit vague and nonspecific. The discussion might benefit from more specific justification, particularly when contextualizing with existing work describing how to design and improve this type of surveillance on practical terms.

We have followed the Reviewer's suggestion and added in lines 311-318 (revised manuscript) the sentence

-- The Alpha experience shows the importance of designing sequencing protocols able to balance sustainability and detection capacity by meeting the minimal requirements of sequencing extent and reporting delay - e.g. sequencing 0.5% of cases with a turnaround time smaller than 21 days as previously proposed (23) -, and by leveraging information from multiple sources, including wastewater and animal surveillance (56,57). Importantly, this study also highlights that the knowledge of surveillance extent and protocol adopted by countries is key to real-time data analysis to better assist risk assessment and intervention planning. This would be facilitated by the widespread adoption of pre-established surveillance protocols.

Line 249. This statement, while true, resonates with some of the problems stated in the beginning of the review. Retrospectively evaluating the need of detection of a specific strain that later on becomes relevant is straightforward. However, if other sources of evidence beyond case surveillance are not available on time, public health bodies will still struggle to make complex decisions. That feels that has to be somehow addressed, otherwise becomes a simplified interpretation and not so useful if the work intends to improve response in the future.

Some of the elements raised by the Reviewer were present in the third paragraph of the Discussion of the original version of the manuscript. We now further expanded it to better highlight the complexity of the various elements at play to take a public health decision. See the reply to the point below.

Line 301. I believe this conclusion from the work is very useful and merits more discussion, as provides evidence that even when an emerging pathogen is not reported, the fact that is not circulating is not necessarily the explanation, but likely it is due to surveillance systems not being able to detect it, specially when major known drivers are indeed in place (i.e., travel flow etc).

We thank the Reviewer for raising this and we borrowed their formulation to further expand the third paragraph of the Discussion where we addressed this issue.

The third paragraph now reads:

-- According to our model, Alpha was introduced in more than 60 countries before the international alert. This provides evidence that when an emerging pathogen is not reported in a given destination country, it may likely be due to the surveillance system not yet being able to detect it. The alert triggered heightened genomic surveillance worldwide, reinstated lockdown measures in the UK, and resulted in border screening and travel bans in countries connected to the UK (23,28–30). However, international response arrived at a moment in which Alpha was already widespread in several countries, preventing containment. Improving surveillance across countries would reduce the time from importation to detection, but it would still clash with the delay needed to recognize a novel variant as a VOC. A lineage with important mutations can be identified relatively quickly if sequencing coverage is high enough (23,24,27), although the assessment of clinical risk is slower (24). Lineages have shown the ability to become dominant without any increase in fitness in particular epidemiological contexts (52), while others like Beta remained at low frequency despite mutations of clinical importance. A more rapid recognition of Alpha as a VOC could have advanced the response by health authorities to delay the establishment of Alpha during a time when vaccination became available in some countries (21). Similar delays in declaring a VOC were also observed for subsequent VOC episodes (19). This underlines the complexity of the interpretation of a context with emerging new variants (53) - especially when major known drivers such as international travel are in place - and of the decision-making for public health response.

REVIEWERS' COMMENTS

Reviewer #1 (Remarks to the Author):

Author has addressed all my previous questions.

Reviewer #2 (Remarks to the Author):

This is a very comprehensive response that answers comments made by me and other reviewers. One minor comment is that I believe the authors should have provided some citations with examples in their response to reviewers letter when they refer to what is a "standard" approach" (page 3).

Reviewer #3 (Remarks to the Author):

The authors have meticulously examined all queries raised by the referees, including those from this reviewer, and have written comprehensive responses, justifications, and, when necessary, additional analyses. Furthermore, the limitations of the approach have been more effectively addressed. This reviewer has no further comments and acknowledges the work to be of a commendable standard.

Reviewer #3 (Remarks on code availability):

I have been able to easily implement and reviewed the code, although not in deep detail

Point-by-point reply of Drivers and impact of the early silent invasion of SARS-CoV-2 Alpha

We thank the Reviewers for their assessment and positive feedback. We provide here below the response to the additional comment of Reviewer 2

Answers to the comments of Reviewer 2

This is a very comprehensive response that answers comments made by me and other reviewers. One minor comment is that I believe the authors should have provided some citations with examples in their response to reviewers letter when they refer to what is a “standard” approach” (page 3).

We thank the Reviewer for the comment and provide below the missing citations.

- Reference for the sentence “exponential growth is a very standard description of epidemic growth”:

Keeling and Rohani, “Modeling Infectious Diseases in Humans and Animals”, Princeton University Press, Sep 19, 2007

- Reference for sentence “proportionality of importations to air traffic is the standard assumption to study international spread of diseases”:

For an early reference see

I. M. Longini et al, Predicting the global spread of new infectious agents, Am J Epidemiol. 1986;123(3):383-91. <https://doi.org/10.1093/oxfordjournals.aje.a114253>.

A similar assumption was used by several works cited in the manuscript, e.g. citations 4-7, 11, 14, 17, 19, 24, 34, 35, 47-51, 55 (revised manuscript)

- Reference for the sentence “ “time to event” models are standard statistical models for rate based descriptions.”:

N.G. Becker, Analysis of Infectious Disease Data, Chapman and Hall/CRC, New-York
DOI <https://doi.org/10.1201/9781315137407>;